# Thinking in Structures: Evaluating Spatial Intelligence in Constraint-Governed Spaces

**Chen Yang** [1]  **Guanxin Lin** [1]  **Youquan He** [1]  **Peiyao Chen** [1]  **Guanghe Liu** [1]  **Yufan Mo** [1]  **Zhouyuan Xu** [1]
**Linhao Wang** [1]  **Guohui Zhang** [1]  **Zihang Zhang** [1]  **Shenxiang Zeng** [1]  **Chen Wang** [1]  **Jiansheng Fan** [1]

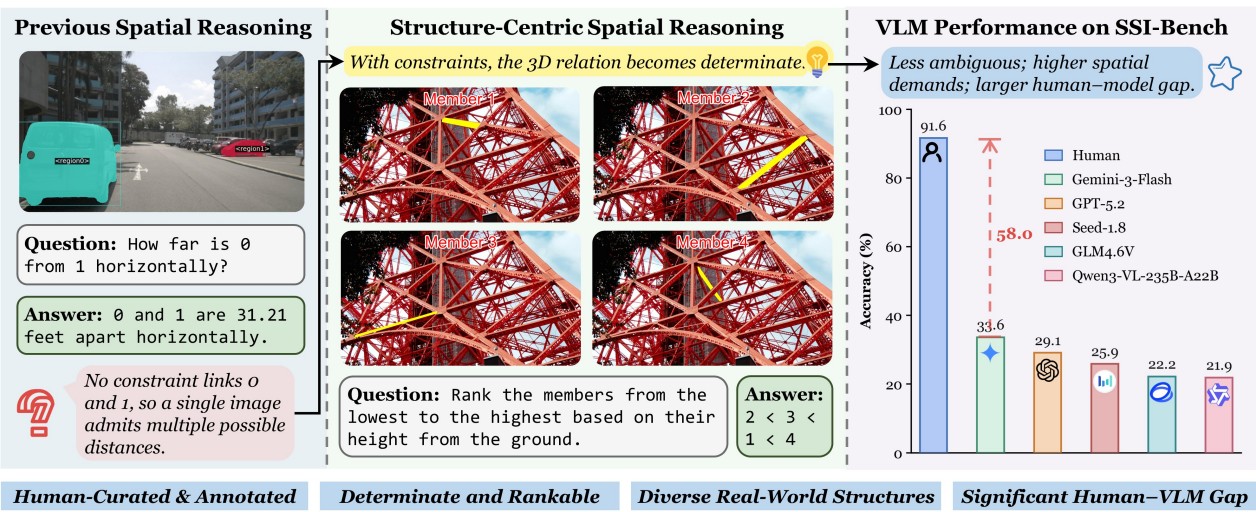

*Figure 1.* **SSI-Bench** is a VQA benchmark designed to evaluate models' spatial reasoning under structural constraints on complex real-world 3D scenes. The bar chart illustrates the significant performance gap between state-of-the-art VLMs and human performance.

## Abstract

Spatial intelligence is crucial for vision–language models (VLMs), yet many scene-centric benchmarks evaluate unconstrained environments where a single image may admit multiple plausible 3D interpretations. We introduce **SSI-Bench**, a VQA benchmark for *Structure-Centric Spatial Reasoning (SCSR)* in constraint-governed spaces. Built from complex real-world 3D structures, it uses structural constraints from geometry, topology, and physical feasibility to make component relations more determinate from visual evidence. The benchmark contains 1,000 ranking questions spanning geometric and topological reasoning, where correct ordering requires resolving all candidate-wise 3D relations, imposing stronger demands on spatial understanding. It

is created through a fully human-centered pipeline with over 400 researcher-hours of image curation, component annotation, and question design. Evaluating 31 VLMs reveals a large gap to humans: the best open-source model achieves 22.2% accuracy and the strongest closed-source model reaches 33.6%, while humans score 91.6%. Further results show that chain-of-thought reasoning brings only marginal gains, and error analysis reveals fundamental limitations in current models' spatial understanding within constraint-governed spaces. Project page: https://ssi-bench.github.io.

## 1. Introduction

Vision–language models (VLMs) have made rapid progress in multimodal understanding and reasoning (Hong et al., 2026; Bai et al., 2025; Li et al., 2025c; Liu et al., 2024). For real-world deployment, however, a central open problem is spatial intelligence: inferring 3D relations and latent structure from visual input and using them to answer geometric and relational queries (Yang et al., 2025a; Yin et al., 2025). Recent benchmarks have expanded along multiple

---
[1]Tsinghua University. Correspondence to: Chen Wang <chwang@tsinghua.edu.cn>, Jiansheng Fan <fan-jsh@tsinghua.edu.cn>.

*Proceedings of the 43rd International Conference on Machine Learning*, Seoul, South Korea. PMLR 306, 2026. Copyright 2026 by the author(s).

axes, including single-view vs. multi-view inference (Cheng et al., 2024; Yang et al., 2025b), images vs. videos (Ma et al., 2025; Lin et al., 2025b), and automated vs. human annotation (Li et al., 2025b; Lin et al., 2025a). These efforts have been valuable for measuring progress, but they provide limited resolution on constraint-governed spaces, where 3D configurations are restricted by structure and feasibility.

We study this regime through *Structure-Centric Spatial Reasoning (SCSR)*: spatial reasoning in which the underlying 3D state is inferred from structural elements and restricted by structural constraints, such as geometric regularities, topological connectivity, and physical feasibility. This perspective reveals an evaluation gap in existing benchmarks. Most spatial benchmarks focus on scene-centric reasoning in largely unconstrained environments, such as indoor navigation and everyday object arrangements, where object configurations are weakly governed by feasibility constraints and can vary almost arbitrarily (Yang et al., 2025a). In single-image settings, this makes many 3D relations underdetermined. For example, an object may appear smaller because it is physically smaller or because it is farther away. Multiple 3D configurations can therefore remain consistent with the same 2D observation, making strict relational queries depend on assumptions, appearance priors, or dataset-specific regularities rather than a uniquely recoverable 3D state.

To address this limitation, we introduce **SSI-Bench** (**S**tructure-centric **S**patial **I**ntelligence Benchmark, Figure 1), a VQA benchmark designed to evaluate SCSR on complex real-world 3D structures. Such structures instantiate constraint-governed spaces: their components follow geometric regularities and connectivity rules, while their realizability is further restricted by physics-based feasibility. These structural constraints reduce single-image ambiguity by making component relations more determinate from visual evidence, enabling precise ranking questions over 3D geometric and topological criteria. Solving these problems requires constraint-consistent 3D understanding under viewpoint variation, clutter, and self-occlusion, together with spatial operations such as mental rotation, cross-sectional inference, occlusion reasoning, and force-path reasoning (Slim et al., 2025; Collins et al., 2022; Chen et al., 2021).

Importantly, this benchmark is not designed to test domain-specific engineering expertise, but to serve as a structure-centric complement to existing scene-centric benchmarks. Its images come from everyday photography sources and cover common real-world structures such as roofs, stairs, bridges, towers, frames, and railings, rather than specialized industrial blueprints. This focus isolates a core aspect of spatial intelligence that is often entangled with object semantics and layout priors in unconstrained scenes. Thus, it complements existing evaluations by testing whether models can recover and reason over coherent 3D structure from real-world visual observations.

Because structural constraints make candidate relations more determinate, ranking provides a well-defined way to evaluate whether models can recover the relative 3D relations required by an explicit geometric or topological criterion. The benchmark contains 1,000 multiple-choice ranking questions, each presenting 3 or 4 candidates, namely members or groups, and requiring selection of the correct permutation. Unlike binary or standard multiple-choice formats, correct ranking requires resolving all relative 3D relations among the candidates, imposing stronger demands on spatial understanding. We organize tasks into two families. Geometric tasks include Ground Height, Ground Angle, Dimension, Relative Distance, Area, and Volume, while Topological tasks evaluate graph-based relations such as hop distance and cycle length. We also include a Multi-View subset to test cross-view correspondence relative to a reference member.

Constructing a diverse real-world benchmark for SCSR is challenging because existing structural datasets rarely provide the explicit spatial metadata needed for automated question generation (Yang et al., 2025a; Lin et al., 2025b; Li et al., 2025b). We therefore develop the benchmark through a fully human-centered pipeline. Ten researchers devoted over 400 hours to reviewing approximately 20,000 structure-related images from multiple sources and selecting over 2,000 candidates that collectively cover nearly all common structure forms, including but not limited to space frames (Xu et al., 2023), steel towers (Bezas et al., 2022), cable-stayed bridges (Qi et al., 2024), timber trusses (Völlmecke et al., 2025), reinforcement frameworks, and pipeline systems (Yang et al., 2022). Candidate sets are curated so that the correct ordering is not reliably recoverable from simple 2D pixel rankings and remains unambiguous under the intended structural interpretation. Each question is further checked by independent reviewers to ensure unambiguity and appropriate challenge.

We evaluate 31 widely used VLMs on the benchmark. The best open-source model achieves 22.2% accuracy and the strongest closed-source model reaches 33.6%, while humans score 91.6%. Chain-of-thought reasoning improves performance only marginally, and error analysis suggests that the gap is primarily driven by limitations in structural grounding and constraint-consistent 3D reasoning.

Overall, our contributions are threefold. First, we introduce **SSI-Bench**, a human-curated benchmark for SCSR in constraint-governed spaces via ranking-based geometric and topological tasks. Second, we evaluate 31 widely used VLMs and human performance, revealing a substantial gap. Third, we provide an error analysis that identifies dominant failure modes and suggests directions for improving structure-centric spatial reasoning.

*Table 1.* Task taxonomy of structure-centric spatial reasoning in `SSI-Bench`.

| Category | Sub-Category | Criterion | Candidates |
|---|---|---|---|
| Geometric | Ground Height | Rank members by centroid height relative to the ground plane. | 4 members |
| | Ground Angle | Rank members by the angle between their principal direction and the ground plane. | 4 members |
| | Dimension | Rank members by their length along the principal direction. | 4 members |
| | Relative Distance | Rank member groups by the minimum distance between their principal-axis lines. | 3 groups |
| | Area | Rank node groups by planar convex-hull area in the reference plane. | 3 groups |
| | Volume | Rank node groups by 3D convex-hull volume enclosing the nodes. | 3 groups |
| | Multi-View | Fuse two views to rank target members by geometric relations to a reference member. | 3 groups |
| Topological | Hop Distance | Rank member groups by shortest-path hop count on the structural connectivity graph. | 3 groups |
| | Cycle Length | Rank member groups by the minimum cycle length with the specified members. | 3 groups |
| | Multi-View | Fuse two views to rank target members by topological relations to a reference member. | 3 groups |

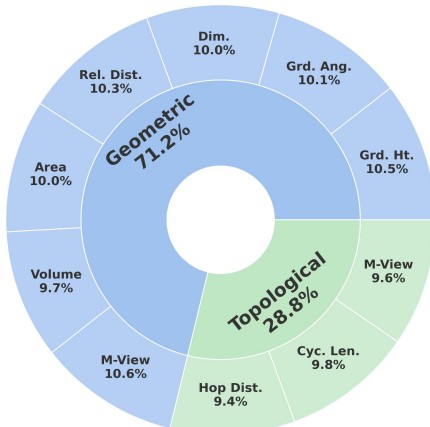

*Figure 2.* Distribution of task categories in `SSI-Bench`.

## 2. Related Work

**Spatial Intelligence Benchmarks.** Benchmarks for spatial intelligence in VLMs have developed along several complementary directions. Early benchmarks are primarily single-view and image-based, emphasizing local metric cues such as depth and distance, *e.g.*, SpatialRGPT (Cheng et al., 2024) and SpatialVLM (Chen et al., 2024). Video-based benchmarks extend evaluation to spatio-temporal understanding with object–object and object–camera relations, including VSI-Bench (Yang et al., 2025a), SPAR-Bench (Zhang et al., 2025a), OST-Bench (Lin et al., 2025b), and MMSI-Video-Bench (Lin et al., 2025a). More recent efforts increasingly emphasize multi-view or multi-image inference to recover 3D structure, as in MMSI-Bench (Yang et al., 2025b), ViewSpatial-Bench (Li et al., 2025a), and Mind-Cube (Yin et al., 2025). Another growing line targets dynamic spatial intelligence by stressing motion, trajectories, and evolving instance states, exemplified by STI-Bench (Li et al., 2025b) and DSI-Bench (Zhang et al., 2025b). Beyond modality and task format, spatial benchmarks can also be viewed along an axis between ecological breadth and reasoning purity. At one end, benchmarks such as VSI-Bench and MMSI-Bench target broad spatial needs in everyday scenes and are closely aligned with embodied AI applications. At the other end, benchmarks such as Spatial457 (Wang et al.,

2025c) and parts of OmniSpatial (Jia et al., 2025) emphasize more diagnostic spatial reasoning with simplified shapes or synthetic objects. Despite this progress, most benchmarks either focus on unconstrained scene-centric settings, where single-image 3D relations can be underdetermined, or on abstract settings that reduce real-world visual complexity. This leaves a complementary gap for benchmarks that retain real-world visual complexity while making 3D relations more determinate through geometric, topological, and physical constraints. Such benchmarks can support precise ranking-based evaluation of structure-centric spatial reasoning in constraint-governed spaces. Detailed quantitative comparisons are provided in Appendix B.

**Structural Understanding Benchmarks.** Structural understanding is commonly evaluated via benchmarks that emphasize part structure, precise geometry, or explicit geometric reasoning. For part-centric structure, PartNet (Mo et al., 2019) provides large-scale 3D objects with fine-grained hierarchical part annotations. 3DCoMPaT++ (Slim et al., 2025) further supports part-based and compositional learning with part-instance-level supervision in a multimodal setting. For geometry-centric structure, ABC (Koch et al., 2019) offers large CAD models with analytic representations and rich ground truth, and datasets such as ABO (Collins et al., 2022) provide real product assets for studying object geometry with multimodal signals. Beyond object datasets, GeoQA (Chen et al., 2021) and VQA benchmarks such as CLEVR (Johnson et al., 2017) and GQA (Hudson & Manning, 2019) evaluate structural reasoning through diagrammatic geometry problems or structured compositional queries, where CLEVR is synthetic and GQA is grounded in real images. Compared with these benchmarks, which often assess structure understanding explicitly via part labels or geometric outputs, our benchmark targets structural understanding implicitly from the perspective of spatial intelligence: models are not asked to output structure directly, but to answer spatial-relation queries that require recovering the underlying 3D structure. It further focuses on real-world 3D structures, introducing richer geometric complexity, occlusion, clutter, and viewpoint variation than curated or simulated settings.

**Geometric**

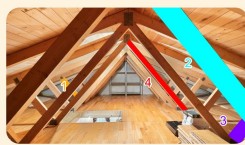

**Ground Height**

**Q:** Rank the members based on their height from the ground (lowest to highest).
**A:** 1, 3, 2, 4

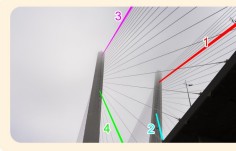

**Ground Angle**

**Q:** Rank the members based on the angle to the ground plane (smallest to largest).
**A:** 3, 1, 4, 2

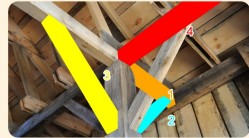

**Dimension**

**Q:** Rank the members based on their length along the main direction (shortest to longest).
**A:** 1, 2, 3, 4

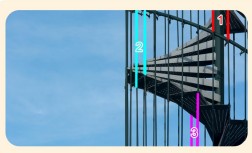

**Relative Distance**

**Q:** Rank the member groups based on their minimum axis-aligned distance (closest to farthest).
**A:** 1, 2, 3

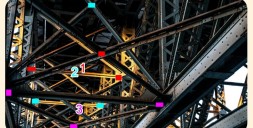

**Area**

**Q:** Rank the node groups based on their planar convex-hull area (smallest to largest).
**A:** 3, 1, 2

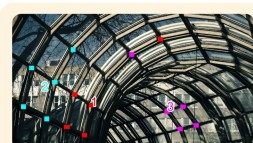

**Volume**

**Q:** Rank the node groups based on their 3D convex-hull volume (smallest to largest).
**A:** 2, 1, 3

**Topological**

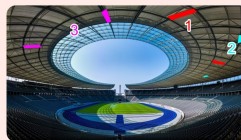

**Hop Distance**

**Q:** Rank the member groups based on the shortest-path hop count (smallest to largest).
**A:** 3, 1, 2

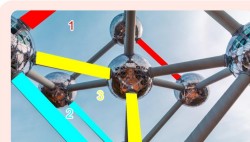

**Cycle Length**

**Q:** Rank the member groups based on the minimum cycle length (shortest to longest).
**A:** 3, 2, 1

**Multi-View**

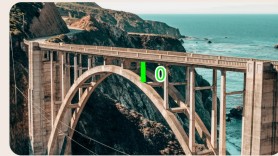 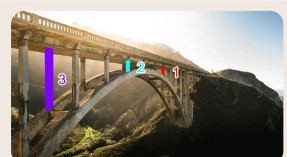

**Multi-View (Geo.)**

**Q:** Rank member 1–3 by their minimum axis-line distance to member 0 (closest to farthest) using two views.
**A:** 2, 3, 1

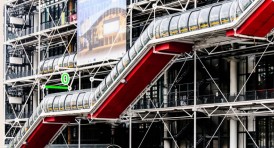 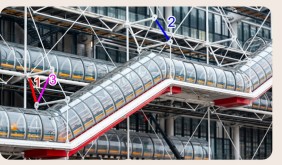

**Multi-View (Topo.)**

**Q:** Rank member 1–3 by their minimum cycle length with member 0 (shortest to longest) using two views.
**A:** 3, 1, 2

*Figure 3.* Representative `SSI-Bench` samples from each category. For visualization, we overlay all candidates in one image; the benchmark provides separately annotated images per option. Ties use the smaller index first. Full questions are in Appendix G.

## 3. `SSI-Bench`

This section introduces `SSI-Bench`, a benchmark for evaluating spatial intelligence under structural constraints. In this regime, solving visual questions requires recovering an underlying 3D geometric and relational structure that is consistent with strong feasibility constraints. We begin by formalizing SCSR in Section 3.1, then outline the benchmark design and task taxonomy in Section 3.2. Section 3.3 details the benchmark construction pipeline.

### 3.1. Problem Formulation

SCSR studies spatial reasoning in which the latent 3D state is restricted to a feasible set defined by explicit constraints.

We represent a structural scene as

$$\mathbf{s} = (V, E, \mathbf{G}, \mathbf{A}), \tag{1}$$

where $V$ and $E$ denote nodes and members, with connectivity graph $\mathcal{G} = (V, E)$. The variable $\mathbf{G}$ collects geometric degrees of freedom (*e.g.*, node coordinates, member directions/lengths, or other latent 3D parameters), and $\mathbf{A}$ captures discrete attributes (*e.g.*, component types or grouping metadata). Feasibility constraints restrict admissible states to a constrained manifold—or more generally, a constrained feasible set—given by

$$\mathcal{M} = \left\{ \mathbf{s} : \mathbf{c}(\mathbf{s}) = \mathbf{0}, \ \mathbf{h}(\mathbf{s}) \le \mathbf{0} \right\}, \tag{2}$$

where $\mathbf{c}$ encodes equality constraints (*e.g.*, geometric compatibility or connectivity-induced relations) and $\mathbf{h}$ encodes

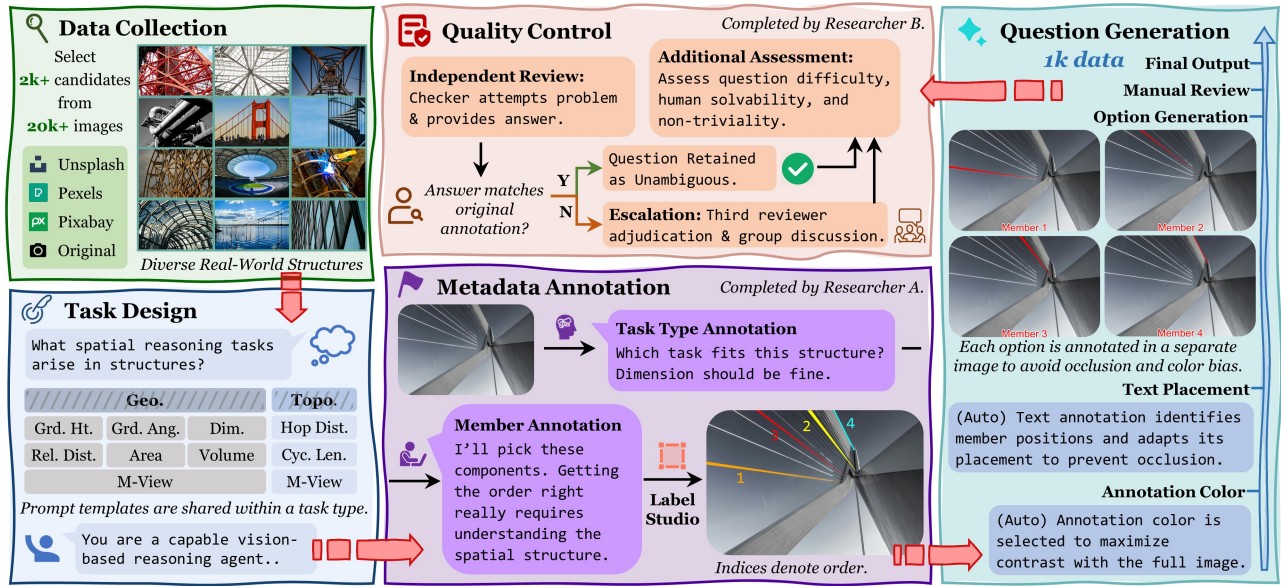

*Figure 4.* Illustration of the **SSI-Bench** construction pipeline.

inequality constraints (*e.g.*, non-intersection, support conditions, or physics-based feasibility). Given an observation $x$ (one or more images), SCSR focuses on inferring relations that remain consistent with both the evidence in $x$ and the feasible set $\mathcal{M}$, rather than relying on unconstrained 2D correlations.

**Instantiation of structural constraints.** In **SSI-Bench**, the structural constraints in Eq. (2) are instantiated by three categories: *geometric regularity*, *topological and connectivity constraints*, and *physical and semantic feasibility*. Geometric regularity and connectivity mainly define equality or discrete compatibility constraints in $\mathbf{c}(\mathbf{s})$ and the graph structure $\mathcal{G}$, while physical and semantic feasibility is naturally captured by inequality constraints in $\mathbf{h}(\mathbf{s})$. These constraints are not provided to models as symbolic inputs; instead, they are implicit in the visual scene and are used during benchmark construction to ensure that the queried 3D relations are well-defined and visually inferable. Further details are provided in Appendix C.

**Ranking formulation.** Each **SSI-Bench** instance specifies a task type $\tau$ and a candidate set $\mathcal{C} = \{c_i\}_{i=1}^K$ (members or groups, with $K \in \{3, 4\}$), together with a criterion function $f_\tau(\mathbf{s}, c)$ that assigns a task-relevant value to each candidate. The ground-truth answer is the permutation $\pi^\star$ that orders candidates by this task-defined value:

$$\pi^\star = \underset{\pi \in S_K}{\operatorname{argsort}} \left( f_\tau(\mathbf{s}, c_{\pi(1)}), \, \dots, \, f_\tau(\mathbf{s}, c_{\pi(K)}) \right). \quad (3)$$

### 3.2. Overview of **SSI-Bench**

Real-world spatial reasoning often hinges on structure. Geometric regularities and connectivity rules restrict how com-

ponents can be arranged, making structural scenes representative constraint-governed spaces. **SSI-Bench** is designed to evaluate SCSR in such spaces.

**SSI-Bench** adopts a structural view of each scene. Each instance is defined by nodes and members as primitives, their geometric attributes, and their topological organization captured by a connectivity graph. Building on this representation, we design two task families—Geometric and Topological—and instantiate them as ranking questions over members or groups. We also include a Multi-View setting that fuses two views to infer relations relative to a reference member and to test cross-view structural consistency. Table 1 and Figure 2 provide a compact overview of the task taxonomy and category distribution. Representative examples from all categories are shown in Figure 3.

The benchmark contains 1,000 multiple-choice ranking questions. Each question presents 3 or 4 candidates and asks for the correct permutation under a specified criterion. Every instance includes a detailed prompt and localization annotations for the referenced members or groups, and is curated to have a unique correct answer with deterministic tie-breaking when necessary. More detailed dataset statistics are reported in Appendix D.

### 3.3. Benchmark Construction Process

Figure 4 summarizes our benchmark construction pipeline. **SSI-Bench** is built through a fully human-centered process: ten researchers with interdisciplinary expertise in AI and structural engineering devoted over 400 hours to its construction. Additional details are provided in Appendix E.

**Data collection.** To target spatial reasoning under structural constraints, we curate real-world images of 3D structures with strong geometric and topological constraints. Most images come from three royalty-free sources—Unsplash (Unsplash), Pexels (Pexels), and Pixabay (Pixabay)—and are filtered for diversity in structure types, scenes, illumination, and viewpoints. After reviewing roughly 20,000 structure-related images, we retain over 2,000 candidates. Because multi-view structural imagery is scarce, part of the Multi-View subset is sourced from our own photography. All selected images are high-resolution and uniformly compressed so that the longer side is at most 1920 pixels.

**Task design.** We define ten task categories across two families, Geometric and Topological, guided by the spatial judgments that naturally arise in complex structures under geometric and connectivity constraints. Ground Height, Ground Angle, and Dimension are member-level tasks with four candidates, while the remaining tasks are group-level with three candidates. The Multi-View subset provides two images per question: one highlights a reference member (Member 0) and the other highlights target members, requiring cross-view correspondence. For each task, we use a task-specific prompt template aligned with its criterion and generate options by permuting candidates into multiple-choice permutations. When ties occur, we apply a consistent ordering rule (placing the smaller index first) to avoid ambiguity.

**Metadata annotation.** We annotate the information needed for ranking using Label Studio. Annotators record an ascending order under the specified criterion and explicitly mark ties. We also provide localization annotations for referenced members or groups. Highlighting polygons are drawn to tightly fit target components while respecting occlusion. Candidate sets are curated so that correct ordering is not reliably recoverable from 2D pixel layouts alone, but instead requires reasoning about the underlying 3D geometry and topology constraints.

**Question generation.** Using the annotated metadata, we instantiate ranking questions with the corresponding images and prompts. To reduce occlusion and color-induced bias, we provide a separately annotated image for each option rather than marking all options in a single image. Highlight colors are automatically selected from a predefined palette to maintain contrast, and text labels are placed near targets with adaptive positioning and then verified by human reviewers for legibility. This process yields 1,000 questions. The metadata annotation and question generation stages can be extended to support semi-automatic data scaling by expanding candidate annotations and sampling new ranking questions, as detailed in Appendix E.6.

**Quality control.** Each question is independently reviewed by a checker who attempts the problem. Questions with

disagreements are escalated to a third reviewer for adjudication. Reviewers also assess whether questions are human-solvable yet challenging, and whether at least one option is non-trivial (i.e., not inferable from superficial 2D cues). We additionally assign a difficulty label to every question. Further details are provided in Appendix E.7.

## 4. Experiments and Analysis

### 4.1. Evaluation Settings

**VLM evaluation.** We evaluate `SSI-Bench` on 31 VLMs, including 10 proprietary models from four families and 21 open-source models from six families. All models are run with temperature 0. Following Enact (Wang et al., 2025a), input images are resized so that the longer side is at most 512 pixels, and we use a unified prompt template for each QA type. Models are instructed to output a parsable Python list encoding a permutation of indices. We report two complementary metrics: *Taskwise Accuracy* (exact-match accuracy on the full permutation) and *Pairwise Accuracy* (pairwise ordering consistency).

**Human evaluation.** We recruit six independent evaluators with basic structural knowledge who were not involved in data annotation. They answer the entire benchmark under the same instructions as the models. Their average performance serves as a proxy for human-level capability on `SSI-Bench`.

**Random baseline.** We report the expected accuracy of uniformly random ranking. Implementation details and metric definitions are provided in Appendices F.1–F.4.

### 4.2. Main Results

Table 2 presents the primary results on `SSI-Bench` in terms of Taskwise Accuracy. We summarize the main findings below. More results are shown in Appendices F.5 and F.6.

**Current VLMs struggle with SCSR.** `SSI-Bench` yields substantially lower accuracies than prior spatial benchmarks in comparable modalities but largely unconstrained settings, indicating that SCSR is harder and less amenable to 2D shortcut cues. Even strong VLMs remain far from human performance. The best proprietary model, Gemini-3-Flash, reaches 33.60% average Taskwise Accuracy, and the best open-source model, GLM-4.6V, reaches 22.20%, while humans achieve 91.60%. The random-ranking baseline is 12.85%, and several models remain close to this level, suggesting that `SSI-Bench` cannot be solved reliably by weak heuristics and that robust SCSR is still missing in today's VLMs.

**Advanced open-source models still trail proprietary counterparts.** A consistent gap appears between open-

*Table 2.* Evaluation on `SSI-Bench` (Taskwise Accuracy). We highlight the best and second best results within each category (Proprietary or Open-source Models). Pairwise Accuracy results are reported in Appendix F.6. Abbreviations: Grd. Ht. = Ground Height; Grd. Ang. = Ground Angle; Dim. = Dimension; Rel. Dist. = Relative Distance; Area = Area; Volume = Volume; M-View = Multi-View; Hop Dist. = Hop Distance; Cyc. Len. = Cycle Length.

| MODELS | GEOMETRIC | | | | | | | TOPOLOGICAL | | | AVG. |
|---|---|---|---|---|---|---|---|---|---|---|---|
| | Grd. Ht. | Grd. Ang. | Dim. | Rel. Dist. | Area | Volume | M-View | Hop Dist. | Cyc. Len. | M-View | |
| *Proprietary Models* | | | | | | | | | | | |
| GEMINI-3-PRO | 25.71 | 37.62 | 28.00 | 33.01 | 24.00 | 27.84 | 31.13 | 35.11 | 30.61 | 21.88 | 29.50 |
| GEMINI-3-FLASH | 37.14 | 38.61 | 35.00 | 41.75 | 27.00 | 25.77 | 33.96 | 32.98 | 34.69 | 28.13 | 33.60 |
| GEMINI-2.5-PRO | 20.95 | 31.68 | 23.00 | 33.98 | 19.00 | 22.68 | 31.13 | 22.34 | 27.55 | 28.13 | 26.10 |
| GEMINI-2.5-FLASH | 20.00 | 24.75 | 21.00 | 21.36 | 23.00 | 12.37 | 26.42 | 26.60 | 24.49 | 22.92 | 22.30 |
| GPT-5.2 | 29.52 | 30.69 | 32.00 | 41.75 | 30.00 | 21.65 | 29.25 | 24.47 | 30.61 | 19.79 | 29.10 |
| GPT-5 MINI | 19.05 | 30.69 | 29.00 | 43.69 | 26.00 | 17.53 | 20.75 | 21.28 | 19.39 | 31.25 | 25.90 |
| GPT-4.1 | 17.14 | 16.83 | 25.00 | 21.36 | 30.00 | 22.68 | 16.04 | 23.40 | 25.51 | 27.08 | 22.40 |
| GPT-4O | 19.05 | 20.79 | 20.00 | 26.21 | 29.00 | 26.80 | 17.92 | 17.02 | 22.45 | 27.08 | 22.60 |
| CLAUDE-SONNET-4.5 | 8.57 | 12.87 | 16.00 | 30.10 | 21.00 | 19.59 | 24.53 | 20.21 | 19.39 | 27.08 | 19.90 |
| SEED-1.8 | 19.05 | 24.75 | 22.00 | 37.86 | 19.00 | 24.74 | 24.53 | 24.47 | 33.67 | 29.17 | 25.90 |
| *Open-source Models* | | | | | | | | | | | |
| GLM-4.6V | 9.52 | 21.78 | 16.00 | 30.10 | 28.00 | 21.65 | 26.42 | 25.53 | 20.41 | 22.92 | 22.20 |
| GLM-4.6V-FLASH | 13.33 | 16.83 | 12.00 | 25.24 | 26.00 | 25.77 | 16.98 | 26.60 | 21.43 | 28.13 | 21.10 |
| GLM-4.5V | 17.14 | 23.76 | 17.00 | 28.16 | 25.00 | 15.46 | 19.81 | 20.21 | 23.47 | 23.96 | 21.40 |
| QWEN3-VL-235B-A22B | 13.33 | 16.83 | 22.00 | 26.21 | 26.00 | 20.62 | 21.70 | 29.79 | 19.39 | 23.96 | 21.90 |
| QWEN3-VL-30B-A3B | 5.71 | 7.92 | 12.00 | 28.16 | 29.00 | 17.53 | 22.64 | 28.72 | 28.57 | 27.08 | 20.60 |
| QWEN3-VL-8B | 9.52 | 5.94 | 12.00 | 28.16 | 18.00 | 18.56 | 18.87 | 24.47 | 29.59 | 28.13 | 19.20 |
| QWEN3-VL-4B | 6.67 | 7.92 | 11.00 | 33.01 | 29.00 | 16.49 | 20.75 | 25.53 | 30.61 | 27.08 | 20.70 |
| QWEN3-VL-2B | 5.71 | 7.92 | 11.00 | 28.16 | 35.00 | 16.49 | 18.87 | 28.72 | 29.59 | 11.46 | 19.20 |
| INTERNVL3.5-241B-A28B | 3.81 | 6.93 | 6.00 | 28.16 | 25.00 | 16.49 | 21.70 | 28.72 | 23.47 | 23.96 | 18.30 |
| INTERNVL3.5-30B-A3B | 6.67 | 7.92 | 12.00 | 30.10 | 32.00 | 16.49 | 18.87 | 27.66 | 28.57 | 28.13 | 20.70 |
| INTERNVL3.5-38B | 6.67 | 5.94 | 16.00 | 22.33 | 29.00 | 15.46 | 22.64 | 27.66 | 18.37 | 27.08 | 19.00 |
| INTERNVL3.5-14B | 5.71 | 7.92 | 11.00 | 16.50 | 29.00 | 17.53 | 23.58 | 19.15 | 19.39 | 30.21 | 17.90 |
| INTERNVL3.5-8B | 5.71 | 6.93 | 10.00 | 24.27 | 33.00 | 16.49 | 24.53 | 25.53 | 28.57 | 28.13 | 20.20 |
| INTERNVL3.5-4B | 8.57 | 6.93 | 7.00 | 27.18 | 28.00 | 14.43 | 15.09 | 17.02 | 24.49 | 19.79 | 16.80 |
| INTERNVL3.5-2B | 4.76 | 3.96 | 11.00 | 22.33 | 25.00 | 17.53 | 4.72 | 11.70 | 10.20 | 0.00 | 11.10 |
| LLAMA-4-SCOUT-17B-16E | 9.52 | 19.80 | 17.00 | 20.39 | 29.00 | 16.49 | 21.70 | 21.28 | 23.47 | 28.13 | 20.60 |
| GEMMA-3-27B | 7.62 | 7.92 | 10.00 | 26.21 | 33.00 | 16.49 | 22.64 | 26.60 | 28.57 | 27.08 | 20.50 |
| GEMMA-3-12B | 8.57 | 4.95 | 8.00 | 16.50 | 18.00 | 17.53 | 22.64 | 19.15 | 29.59 | 29.17 | 17.30 |
| GEMMA-3-4B | 6.67 | 6.93 | 12.00 | 22.33 | 32.00 | 18.56 | 23.58 | 26.60 | 29.59 | 19.79 | 19.70 |
| LLAVA-ONEVISION-72B | 4.76 | 6.93 | 4.00 | 20.39 | 31.00 | 16.49 | 23.58 | 19.15 | 23.47 | 22.92 | 17.20 |
| LLAVA-ONEVISION-7B | 9.52 | 5.94 | 3.00 | 19.42 | 19.00 | 19.59 | 24.53 | 23.40 | 22.45 | 18.75 | 16.50 |
| *Baselines* | | | | | | | | | | | |
| RANDOM GUESSING | 4.17 | 4.17 | 4.17 | 16.67 | 16.67 | 16.67 | 16.67 | 16.67 | 16.67 | 16.67 | 12.85 |
| **HUMAN PERFORMANCE** | **94.29** | **91.09** | **92.00** | **93.20** | **98.00** | **91.75** | **89.62** | **92.55** | **89.80** | **83.33** | **91.60** |

source and proprietary models across both geometric and topological tasks. Proprietary systems dominate the top of the leaderboard (*e.g.*, Gemini-3-Flash at 33.60%, Gemini-3-Pro at 29.50%, and GPT-5.2 at 29.10%), whereas leading open-source models remain lower (*e.g.*, GLM-4.6V at 22.20% and Qwen3-VL-235B-A22B at 21.90%). This gap suggests that current open-weight models are less reliable at inferring the latent 3D state and applying structural constraints for correct rankings, while the absolute performance of proprietary models also indicates that SCSR remains broadly unsolved.

**Progress over time is visible but remains incremental, and scaling yields limited and inconsistent gains.** Across major model lineages, performance improves gradually over successive generations, but the gains are uneven and often modest. Within Gemini, average accuracy increases from Gemini-2.5-Flash (22.30%) and Gemini-2.5-Pro (26.10%) to Gemini-3-Pro (29.50%) and Gemini-3-Flash (33.60%). A similar pattern appears in the GPT series, where GPT-4.1 (22.40%) and GPT-4o (22.60%) improve to GPT-5 mini (25.90%) and GPT-5.2 (29.10%). In contrast, progress within GLM is smaller, with GLM-4.5V (21.40%) and GLM-4.6V (22.20%) remaining close. Larger or more recent variants do not always improve substantially within the same family, suggesting that scaling alone is insufficient to close the gap to human-level structural reasoning. These trends point to the need for improvements beyond scaling, such as stronger structure-aware training signals and better

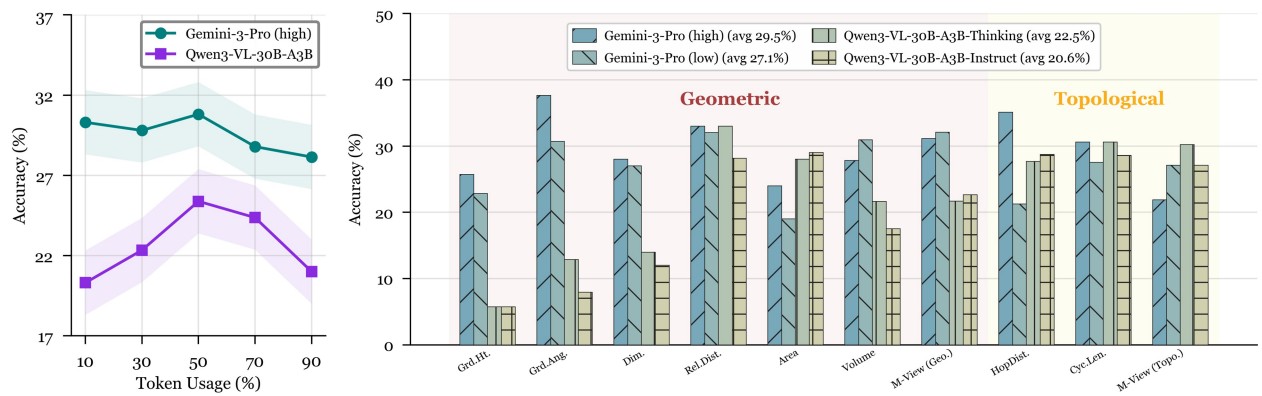

*Figure 5.* (*Left*) Relationship between thinking-token usage and accuracy; (*Right*) Sub-category level effects of thinking on SCSR.

coverage of constrained 3D structural configurations.

## 4.3. Impact of Thinking on SCSR

We study how explicit "thinking" affects SCSR by comparing two representative VLMs under the same evaluation protocol as Section 4.1: Gemini-3-Pro with two thinking levels (HIGH vs. LOW) and Qwen3-VL-30B-A3B with two variants (THINKING vs. INSTRUCT). Each setting is evaluated on the full **SSI-Bench** benchmark. Figure 5 shows the results. Additional details are provided in Appendix F.7.

**Thinking improves performance, but only modestly.** Stronger thinking consistently increases Taskwise Accuracy, yet the gains are small relative to the overall difficulty of SCSR. Gemini-3-Pro improves from 27.1% (LOW) to 29.5% (HIGH), and Qwen3-VL-30B-A3B improves from 20.6% (INSTRUCT) to 22.5% (THINKING). These results indicate that "thinking" provides incremental benefits rather than resolving the dominant failure modes on **SSI-Bench**.

**Token usage is a weak proxy for effective reasoning.** We bin questions by token usage (fraction of the maximum thinking-token count) and compute accuracy per bucket (Figure 5, *Left*). For both models, accuracy is non-monotonic: it peaks at moderate usage and drops at very low or very high usage. This suggests that extra tokens often signal uncertainty rather than improved constraint-consistent inference. High usage frequently corresponds to prolonged deliberation over incorrect structural hypotheses (*e.g.*, mislocalization or cross-view mismatch), which amplifies errors; the decline at the highest usage bucket is more pronounced for Qwen3-VL-30B-A3B, while Gemini-3-Pro remains comparatively stable.

**Benefits are task-dependent and can be negative on 3D-consistency bottlenecks.** Per-task results (Figure 5, *Right*) show that thinking does not yield uniform gains. For Gemini-3-Pro, HIGH thinking improves several single-view geometric/topological criteria (*e.g.*, Ground Angle and HopDist.), but it is mixed—and sometimes worse—on tasks

that hinge on globally consistent 3D reconstruction, particularly the Multi-View setting and Volume. Overall, these patterns suggest that additional deliberation helps mainly when the evidence supporting the task criterion is stable, but it can amplify errors when structural grounding or cross-view correspondence is uncertain, leading to longer reasoning over incorrect 3D hypotheses.

## 4.4. Error Analysis

To diagnose the bottlenecks of current VLMs on SCSR, we perform an error analysis with Gemini-3-Pro as a representative model. Following the evaluation protocol in Section 4.1, we collect its reasoning traces and randomly sample 100 questions from **SSI-Bench** for manual inspection. We identify four dominant failure modes; Figure 6 presents representative examples. More complete cases and analyses are provided in Appendix H.

**Member-extent errors.** The model frequently misestimates the spatial extent of a highlighted member, implicitly extending it beyond its true endpoints or truncating it to a visible fragment. This failure is particularly prevalent under occlusion and clutter: when only a portion of a member is visible, the model often treats the fragment as the entire component. Such extent errors directly corrupt geometric comparisons that rely on correct endpoints and principal directions, leading to unreliable rankings in tasks such as *Dimension* and *Relative Distance*.

**Object-recognition errors.** We observe two recurring patterns. First, the model misidentifies components or nodes (*e.g.*, confusing stair treads with diagonal braces), with errors increasing when targets are small, thin, or heavily occluded. Second, it misjudges coarse orientation, such as interpreting a slanted member as horizontal or a near-vertical member as perfectly vertical. These recognition and orientation failures most directly degrade *Ground Angle*, and they also propagate to downstream comparisons that depend on correct component identity and orientation.

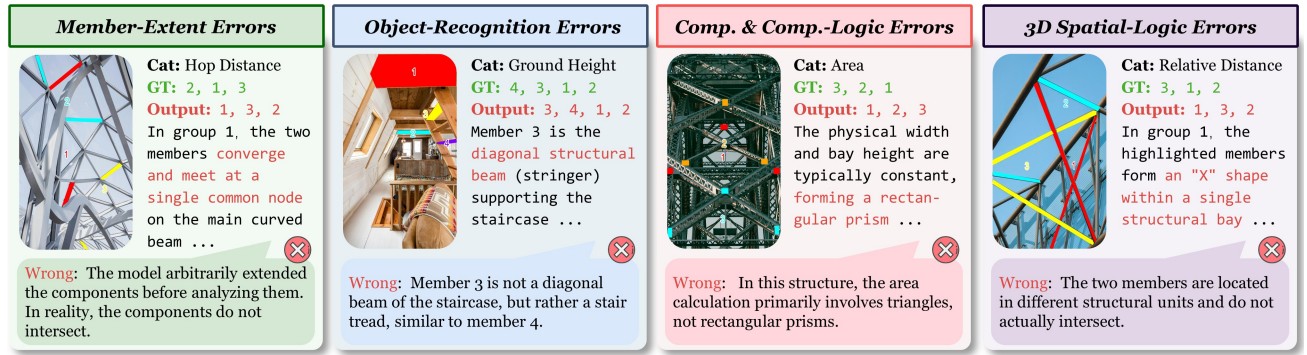

*Figure 6.* Illustration of four error types identified in VLM spatial reasoning on `SSI-Bench`.

**Computational and comparison-logic errors.** Even when the relevant components are localized correctly, the model may apply an incorrect comparison rule. For *Area* and *Volume*, it sometimes optimizes the wrong quantity (*e.g.*, reasoning about 2D projected area when the criterion is 3D convex-hull volume) or adopts invalid simplifications (*e.g.*, substituting an oblique height with an orthogonal height). For length-related criteria, the model occasionally falls back on coarse, experience-based heuristics rather than computing the task-defined relation under the implied 3D structure, producing plausible yet incorrect orderings.

**3D spatial-logic errors.** These failures reflect limitations in reconstructing and reasoning over a globally consistent 3D state under structural constraints. The model often shows weak depth reasoning, confusing near–far relations under perspective and foreshortening (common in cable, truss, and tower scenes). In Multi-View questions, it may fail to establish consistent cross-view correspondences, leading to incompatible interpretations across views. It also sometimes composes relations incorrectly; for example, from "Member 1 is left of Member 2" and "Member 3 is above Member 1," it may wrongly infer "Member 3 is left of Member 2," indicating unstable reference selection and non-robust relational composition.

Overall, these error modes suggest that the `SSI-Bench` performance gap arises not only from imperfect visual grounding, but also from limited 3D structural reconstruction and constraint-consistent spatial inference—capabilities central to SCSR. They also suggest two directions for future improvement: enhancing fine-grained structural perception through captions or annotations of component locations, extents, connectivity, and relations, and strengthening spatial reasoning through large-scale chain-of-thought data for structure-centric tasks.

## 5. Conclusions

In this work, we present `SSI-Bench`, a human-curated benchmark that targets spatial reasoning under structural constraints in vision–language models using complex real-world 3D structures. Evaluations across 31 VLMs expose a pronounced gap between current models and human performance, underscoring the difficulty of recovering constraint-consistent 3D structure from visual input. Our analyses further show that explicit thinking yields only modest improvements, with most errors tracing back to limitations in structural grounding and globally coherent 3D reasoning. We expect `SSI-Bench` to facilitate more fine-grained evaluation and to inform future advances in spatially grounded multimodal systems. Limitations are discussed in Appendix I.

## Acknowledgements

The authors gratefully acknowledge the financial support provided by the National Natural Science Foundation of China (Grant No. 52408188).

## Impact Statement

This work aims to advance the evaluation of spatial intelligence in vision–language models. While improved spatial reasoning can benefit applications such as robotics, accessibility, and infrastructure inspection, it may also have dual-use implications in sensitive contexts. `SSI-Bench` is an offline evaluation benchmark and does not provide action policies, control interfaces, or deployment data. We encourage its responsible use as a diagnostic tool for robust and transparent spatial reasoning.

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

# A. Appendix Overview and Organization

This appendix provides supplementary details to support and extend the main paper, with an emphasis on transparency and reproducibility. The appendix is organized as follows.

**(1) Comparison with existing benchmarks (Section B).** We provide detailed comparisons between `SSI-Bench` and prior benchmarks to clarify the positioning and distinctive challenges covered by our benchmark.

**(2) Structural constraint types (Section C).** We provide a detailed categorization of the structural constraints used in `SSI-Bench`, including geometric regularity, topological and connectivity constraints, and physical and semantic feasibility.

**(3) Dataset statistics (Section D).** We report summary statistics of the dataset and the distributional properties of instances across tasks and categories.

**(4) Details of the Data Curation Process (Section E).** We provide additional details on the benchmark construction process beyond the main text, including data sources (Section E.1), data annotation and quality control interfaces (Section E.2), general annotation guidelines (Section E.3), the data format (Section E.4), tie handling and evaluation fairness (Section E.5), semi-automatic expansion (Section E.6), and the difficulty annotation methodology (Section E.7).

**(5) Additional implementation details and experimental results (Section F).** We provide metric definitions (Section F.1), evaluated models and inference settings (Section F.2), implementation details for model evaluation (Section F.3), and the human evaluation protocol (Section F.4), along with additional results such as pairwise accuracy (Section F.6) and complete analyses of thinking-related settings (Section F.7).

**(6) Representative benchmark examples (Section G).** We present representative instances from each category with full prompts and visual annotations to illustrate the diversity of structure types and task demands.

**(7) Additional case studies (Section H).** We provide further qualitative analyses illustrating typical model behaviors and failure patterns on SCSR, complementing the examples discussed in the main text.

**(8) Limitations (Section I).** We discuss limitations of `SSI-Bench` and our evaluation protocol, and outline directions for future work.

Overall, these sections provide additional transparency on dataset construction and evaluation, enabling reproducibility and broader contextualization of our findings.

# B. Comparison with Other Spatial Intelligence Benchmarks

Table 3 compares `SSI-Bench` with representative spatial intelligence benchmarks along the axes used throughout our paper: (i) regime (unconstrained vs. structurally constrained), (ii) modality (single-/multi-image vs. video/multi-video), (iii) annotation (automatic vs. human), and (iv) task coverage (spatial understanding, motion understanding, and cross-video reasoning). Beyond cataloging differences, we emphasize a central theme of `SSI-Bench`: by operating in a *structurally constrained* regime, SCSR places a *higher demand on genuine 3D understanding*—models must infer a constraint-consistent 3D scene hypothesis rather than rely on weak 2D correlations. At the same time, these strong structural constraints narrow the space of feasible 3D configurations and make the queried relations more stable under plausible interpretations, enabling evaluation that is more *quantifiable and comparable* across models and runs.

**Single-image benchmarks.** Early spatial benchmarks largely evaluate spatial understanding from a single view in everyday environments where feasible 3D configurations are only weakly restricted. SpatialRGPT (Cheng et al., 2024) and SpatialVLM (Chen et al., 2024) focus on local metric or relational cues from one image, typically relying on automatic or mixed auto/human annotations. CVBench (Zhu et al., 2025) further scales single-image evaluation with automatically generated questions. These benchmarks are valuable for tracking progress on local spatial cues, but their scenes are commonly weakly constrained: many different 3D states can plausibly explain the same observation. As a result, models can sometimes achieve non-trivial performance via 2D correlations, appearance priors, or dataset regularities, without committing to a constraint-consistent 3D structural interpretation.

**Multi-image benchmarks.** A second line improves over single-view ambiguity by providing multiple images and probing cross-view reasoning. MultiSPA (Xu et al., 2025) introduces multi-image settings with automatic generation; All-Angles-Bench (Yeh et al., 2025) emphasizes viewpoint changes with human annotations. MMSI-Bench (Yang et al., 2025b), ViewSpatial-Bench (Li et al., 2025a), and MindCube (Yin et al., 2025) expand multi-image evaluation in task diversity and

*Table 3.* Comparison of `SSI-Bench` with representative spatial reasoning benchmarks in terms of evaluation regime, modality, annotation type, and task coverage. SU/MU denote spatial/motion understanding; CV denotes cross-video reasoning. SC/UC indicate structurally constrained/unconstrained regimes. Human–AI Gap is the absolute performance difference (percentage points) between humans and the best reported AI system.

| Benchmark | Regime | Modality | Annotation | Task | Samples | Human–AI Gap |
|---|---|---|---|---|---|---|
| SpatialRGPT (Cheng et al., 2024) | UC | Single-Image | Auto | SU | 1,406 | <42 |
| SpatialVLM (Chen et al., 2024) | UC | Single-Image | Auto & Human | SU | 546 | <30 |
| CVBench (Zhu et al., 2025) | UC | Single-Image | Auto | SU | 2,638 | – |
| MultiSPA (Xu et al., 2025) | UC | Multi-Image | Auto | SU & MU | 7,800 | – |
| All-Angles-Bench (Yeh et al., 2025) | UC | Multi-Image | Human | SU | 2,100 | 21.2 |
| MMSI-Bench (Yang et al., 2025b) | UC | Multi-Image | Human | SU & MU | 1,000 | 55.3 |
| ViewSpatial-Bench (Li et al., 2025a) | UC | Multi-Image | Auto & Human | SU | 5,712 | – |
| MindCube (Yin et al., 2025) | UC | Multi-Image | Auto & Human | SU & MU | 21,154 | – |
| VSI-Bench (Yang et al., 2025a) | UC | Video | Auto | SU | 5,000 | 33 |
| OST-Bench (Lin et al., 2025b) | UC | Video | Auto | SU & MU | 10,000 | 29.3 |
| SPAR-Bench (Zhang et al., 2025a) | UC | Video | Auto | SU & MU | 7,207 | 27.8 |
| STI-Bench (Li et al., 2025b) | UC | Video | Auto | SU & MU | 2,064 | – |
| DSI-Bench (Zhang et al., 2025b) | UC | Video | Human | SU & MU | 1,000 | – |
| EgoExoBench (He et al., 2025) | UC | Multi-Video | Auto & Human | CV | 7,330 | 41.9 |
| MMSI-Video (Lin et al., 2025a) | UC | Video/Multi-Video | Human | SU & MU & CV | 1,106 | 58.4 |
| **SSI-Bench (ours)** | **SC** | **Single-/Multi-Image** | **Human** | **SU** | **1,000** | **58.0** |

scale, aiming to test whether models can fuse observations to infer more reliable 3D relations. While adding views generally strengthens supervision and reduces under-determination, these benchmarks still predominantly operate in unconstrained everyday regimes, where feasible configurations span a broad space. Consequently, even in multi-image settings, models may partially rely on multi-view correlations that do not require enforcing strong feasibility constraints or recovering a globally consistent structural state.

**Video benchmarks for spatio-temporal reasoning.** Video-based benchmarks extend spatial intelligence to spatio-temporal understanding and often include motion-centric tasks. VSI-Bench (Yang et al., 2025a) tests spatio-temporal spatial understanding and planning-related components; OST-Bench (Lin et al., 2025b) targets object–state transitions; SPAR-Bench (Zhang et al., 2025a) evaluates richer spatio-temporal reasoning. STI-Bench (Li et al., 2025b) and DSI-Bench (Zhang et al., 2025b) further emphasize dynamic instance states, with DSI-Bench incorporating human annotation. These efforts broaden evaluation beyond static geometry, but most still focus on scenes with weak feasibility constraints, where strong performance may be attainable through temporal tracking heuristics or action priors, rather than constraint-consistent 3D reconstruction.

**Multi-video and cross-video benchmarks.** More recent benchmarks investigate correspondence and reasoning across videos. EgoExoBench (He et al., 2025) evaluates alignment across egocentric/exocentric views, and MMSI-Video (Lin et al., 2025a) extends multi-video reasoning with human annotation and additional task coverage (e.g., planning, cross-video reasoning). These datasets raise the difficulty through cross-episode generalization, yet their underlying environments are typically not governed by strong geometric/topological feasibility constraints.

**SSI-Bench: structural constraints enable well-defined ranking-based evaluation.** In contrast to the above unconstrained benchmarks, `SSI-Bench` targets SCSR, which places a *higher demand on genuine 3D understanding*: models must recover a constraint-consistent 3D structural hypothesis (geometry *and* topology) from visual evidence and reason over it, rather than relying on weakly constrained 2D correlations or appearance priors. The strong geometric, topological, and physics-based constraints of real-world structures sharply narrow the space of feasible configurations, making many queried relations more stable under plausible interpretations and thereby enabling evaluation that is more *quantifiable and comparable* across models. To leverage this property, `SSI-Bench` adopts a ranking formulation over 3–4 candidates under explicit geometric/topological criteria, which emphasizes robust comparative reasoning and converts constraint-induced stability into a directly measurable signal, especially in cluttered, occluded, and viewpoint-diverse structural scenes.

## C. Structural Constraint Types

We categorize the constraints used in `SSI-Bench` into three groups: geometric regularity, topological and connectivity constraints, and physical and semantic feasibility, as summarized in Table 4. These constraint types are not provided to

*Table 4.* Categorization and description of structural constraints in `SSI-Bench`.

| Category | Constraint Type | Mechanism | Example | 3D Inference Benefit |
|---|---|---|---|---|
| Geometric Regularity | Symmetry | Mirrored geometry and node positions across an axis or plane. | A pitched house roof or a symmetrical stadium dome, where the left and right halves are mirrored. | Helps infer occluded or heavily foreshortened parts from their visible counterparts. |
| | Parallelism & Orthogonality | Members align parallel ($A \parallel B$) or orthogonal ($A \perp B$). | A multi-story parking garage, where floor slabs are parallel and support columns are orthogonal to floors. | Establishes a global coordinate frame to resolve perspective distortion. |
| | Coplanarity & Collinearity | Nodes or members share a single 3D plane or line. | The flat glass curtain wall of a modern commercial building facade. | Restricts depth variation and reduces geometric degrees of freedom. |
| | Periodicity & Equidistance | Structural units repeat with equal or regular spacing. | Bleacher seating in a stadium or regular steps on a public staircase. | Replaces absolute metric measurement with structural counting. |
| Topological & Connectivity | Joint Sharing | Converging members share exact 3D endpoint coordinates. | A steel space-frame roof where multiple struts meet at a single spherical node. | Provides hard equality constraints and prevents inconsistent floating endpoints. |
| | Sequence & Enclosure | Fixed connection rules form ordered chains or closed invariant loops. | The side rails of a pedestrian truss bridge, forming closed and stable triangular patterns. | Preserves invariant topology under 2D projection and narrows feasible 3D layouts. |
| | Anchoring | Base nodes rest on a common datum plane. | The concrete piers of a highway overpass, all anchored to the same ground-level foundation. | Establishes an absolute zero-elevation reference. |
| Physical & Semantic | Gravity Alignment | Load-bearing members align with gravity or remain level relative to it. | The main load-bearing columns of a shopping mall and its horizontal floor decks. | Defines a global vertical axis and resolves up–down ambiguity under camera tilt. |
| | Shape Physics | Components deform into predictable, physics-governed shapes. | The main cables of a large suspension bridge, forming a catenary curve under load. | Injects physical priors to prevent arbitrary or implausible 3D interpretations. |

models during evaluation. Instead, they describe the implicit regularities used during benchmark construction to select suitable scenes, design unambiguous questions, and ensure that the target rankings can be inferred from visual evidence rather than from superficial 2D cues.

Geometric regularity constrains the metric and directional layout of components. Topological and connectivity constraints restrict how nodes and members can be connected and how graph-level relations are preserved under projection. Physical and semantic feasibility rules out implausible configurations according to support, gravity, material behavior, and the functional roles of structural components. Together, these constraints reduce the space of plausible 3D configurations and make the queried spatial relations more stable under reasonable interpretations.

## D. Dataset Statistics

Figure 2 and Table 1 in the main text summarize the category distribution and task taxonomy of `SSI-Bench`. This appendix provides additional statistics on sample counts, image usage, and prompt lengths. In total, the benchmark contains 1,000 multiple-choice ranking questions. Each question compares either member groups or node groups under an explicit geometric or topological criterion and requires selecting the correct permutation among 3 or 4 candidates.

Table 5 reports the number of questions, unique images, and template prompt length for each task category. Across the 1,000 questions, we use 1,160 unique images (#UI), reflecting that the Multi-View setting consumes two views per instance and thus contributes more unique images than single-view categories. The average template prompt length (#TPL) is 2,062 characters, measured after filling in the task-specific template text, including spaces and punctuation. This provides a coarse proxy for linguistic complexity and annotation density.

As shown in the main-text distribution figure, `SSI-Bench` spans two task families: Geometric and Topological. The

*Table 5.* Additional statistics of SCSR tasks in `SSI-Bench`. #UI denotes the number of unique images. #TPL denotes the template prompt length in characters, including spaces and punctuation.

| Category | Sub-Category | Count | #UI | #TPL |
|---|---|---|---|---|
| Geometric | Ground Height | 105 | 105 | 1,662 |
| | Ground Angle | 101 | 101 | 1,892 |
| | Dimension | 100 | 100 | 1,820 |
| | Relative Distance | 103 | 103 | 1,782 |
| | Area | 100 | 100 | 1,467 |
| | Volume | 97 | 97 | 1,489 |
| | Multi-View | 106 | 192 | 2,404 |
| Topological | Hop Distance | 94 | 94 | 2,475 |
| | Cycle Length | 98 | 98 | 2,511 |
| | Multi-View | 96 | 170 | 3,158 / 3,225 |
| **Total / Avg.** | | **1,000** | **1,160** | **2,062** |

Geometric family covers six single-view tasks plus a Multi-View variant, with 712 questions in total. The Topological family contains two single-view graph-based tasks and a Multi-View variant, with 288 questions in total. Overall, the benchmark emphasizes structure-centric geometric comparisons while preserving substantial coverage of connectivity reasoning.

The Multi-View setting is designed to test cross-view correspondence and structural consistency while keeping the queried relation well-defined. For the Geometric family, Multi-View is effectively the multi-view version of Relative Distance: it fuses two viewpoints to rank target members by their geometric relations to a reference member. For the Topological family, Multi-View includes two template variants because it covers both Hop Distance and Cycle Length under multi-view querying. Accordingly, Table 5 reports two #TPL values for this row, corresponding to the two prompt templates used in the Multi-View topological setting. The full prompt templates, including the exact wording for each variant, are provided in Section G.

The #TPL column also shows that Topological tasks typically require longer prompts than Geometric tasks, because they must specify graph-based criteria and precisely identify queried members or groups within potentially cluttered connectivity patterns. Multi-View prompts are generally longer than single-view prompts due to explicit instructions for fusing information from two images and establishing correspondence to the same structural elements across views. These statistics reflect the intended challenge of the benchmark: solving the questions requires reconstructing a constraint-consistent 3D or graph structure rather than relying on simple 2D heuristics.

## E. Details of the Data Curation Process

### E.1. Dataset Sources

**Unsplash.** Unsplash (Unsplash) is a large repository of high-resolution, royalty-free photographs. We use it as a primary source to collect single-view images of real-world 3D structures with salient geometric regularities and viewpoint variation, filtering for diversity in structure types, scenes, illumination, and camera perspectives.

**Pexels.** Pexels (Pexels) provides curated royalty-free images with diverse compositions and environments. We use it to complement Unsplash with additional structural scenes featuring clutter, self-occlusion, and varied viewpoints, and curate candidates to reduce reliance on pixel-level shortcuts.

**Pixabay.** Pixabay (Pixabay) is a royalty-free media library with broad topical coverage. We use it to further increase diversity in structural instances and visual conditions (e.g., materials and lighting), and retain images that highlight constraint-governed 3D structure suitable for SCSR.

**Original Photography (Multi-View Subset).** Because multi-view structural imagery is scarce in public sources, part of our Multi-View subset is captured by us. These paired views support evaluating cross-view correspondence and structural consistency relative to a reference member. Across all sources, selected images are high-resolution and uniformly compressed so that the longer side is at most 1920 pixels.

In total, we collect 1,160 candidate images from four sources: Pexels contributes 500 images (43.1%), Unsplash contributes 474 (40.9%), Pixabay contributes 102 (8.8%), and our original photography contributes 84 (7.2%).

## E.2. Data Annotation and Quality Control Interfaces

To ensure consistent metadata collection and rigorous quality control, we develop three lightweight, human-in-the-loop interfaces corresponding to (i) metadata annotation, (ii) manual review during question generation, and (iii) independent review for quality control. Figures 7–9 show representative screenshots.

**Metadata annotation interface.** Figure 7 shows the interface used for metadata annotation in Label Studio. For each image (or image pair in the Multi-View subset), annotators first specify the task category and then select the referenced entities (members or groups) according to the task definition. The interface supports recording the target ranking as an *ascending order* under the task-specific criterion, and allows annotators to explicitly mark equality relations (ties) among candidates to avoid ambiguous supervision. In addition, annotators provide localization annotations for the referenced members/groups, which are later used to render option-specific highlighted images for question generation.

**Manual review interface for question generation.** After instantiating questions from the annotated metadata, we perform a manual review step to verify the faithfulness and readability of visual annotations. As shown in Figure 8, reviewers inspect each option-specific annotated image to check (1) whether highlighting colors maintain sufficient contrast against the background and other marks, and (2) whether text labels are placed appropriately (i.e., close to the intended targets, non-overlapping, and not occluded). When issues are found (e.g., low-contrast colors, labels covering key geometry, or confusing placement), reviewers revise the rendering configuration (color choice or label position) to ensure that the visualization is clear while avoiding unintended cues.

**Independent review interface for quality control.** Figure 9 illustrates the interface used for independent quality control. For each question, a checker attempts the problem under the same format as the benchmark evaluation and provides their final answer (and, when applicable, notes about ambiguity or potential flaws). Questions with disagreements between the original annotation-derived answer key and the checker's answer are escalated to a third reviewer for adjudication. This interface also supports structured flags for common failure modes (e.g., unclear target identification, excessive occlusion, or options that are trivially inferred from 2D cues), enabling systematic filtering and iterative refinement of the dataset.

## E.3. General Guidelines

We provide the following guidelines to ensure that **SSI-Bench** annotations are accurate, unambiguous, and consistently aligned with the intended evaluation of SCSR.

- **Human-centric design.** All questions and annotations are created to be understandable to humans, using clear natural-language descriptions rather than technical or camera-parameter-based prompts.

- **Answer determinism and uniqueness.** Each instance must admit a *single, well-defined* correct answer. Annotators should actively avoid ambiguous cases (e.g., unclear targets, visually indistinguishable options, or multiple plausible interpretations). This design choice differs from interaction-oriented benchmarks such as InteractComp (Deng et al., 2025).

- **Moderate, meaningful difficulty.** Questions should require careful reasoning by a non-expert human, but remain solvable. Trivial shortcuts that rely on superficial 2D cues should be avoided (e.g., when the correct ordering matches the pixel-height ordering in the image without requiring 3D or structural reasoning).

- **Consistent candidate specification.** For member-level tasks, annotators label four candidates (1–4); for group-level tasks, annotators label three candidates (1–3). Candidate sets should be selected such that the intended comparison is non-trivial yet clearly resolvable.

- **Precise localization and occlusion handling.** Polygons should tightly fit the target components while respecting occlusion. If a component is occluded, annotators should mark the visible parts as separate polygons for the same target when necessary, rather than merging distinct components into one region. If a component is incomplete in the image, annotators default to the smallest identifiable unit that can be treated as an atomic element.

- **Strict ordering and tie annotation.** Annotations must follow the ground-truth ranking order *from smallest to largest* under the specified criterion; annotators should double-check to avoid reversed ordering. If ties exist, annotators explicitly mark equality relations (e.g., selecting 1=2 indicates 1=2<3<4). During question generation, ties are resolved by a deterministic rule: if two candidates are indistinguishable, the smaller index is considered earlier in the ordering.

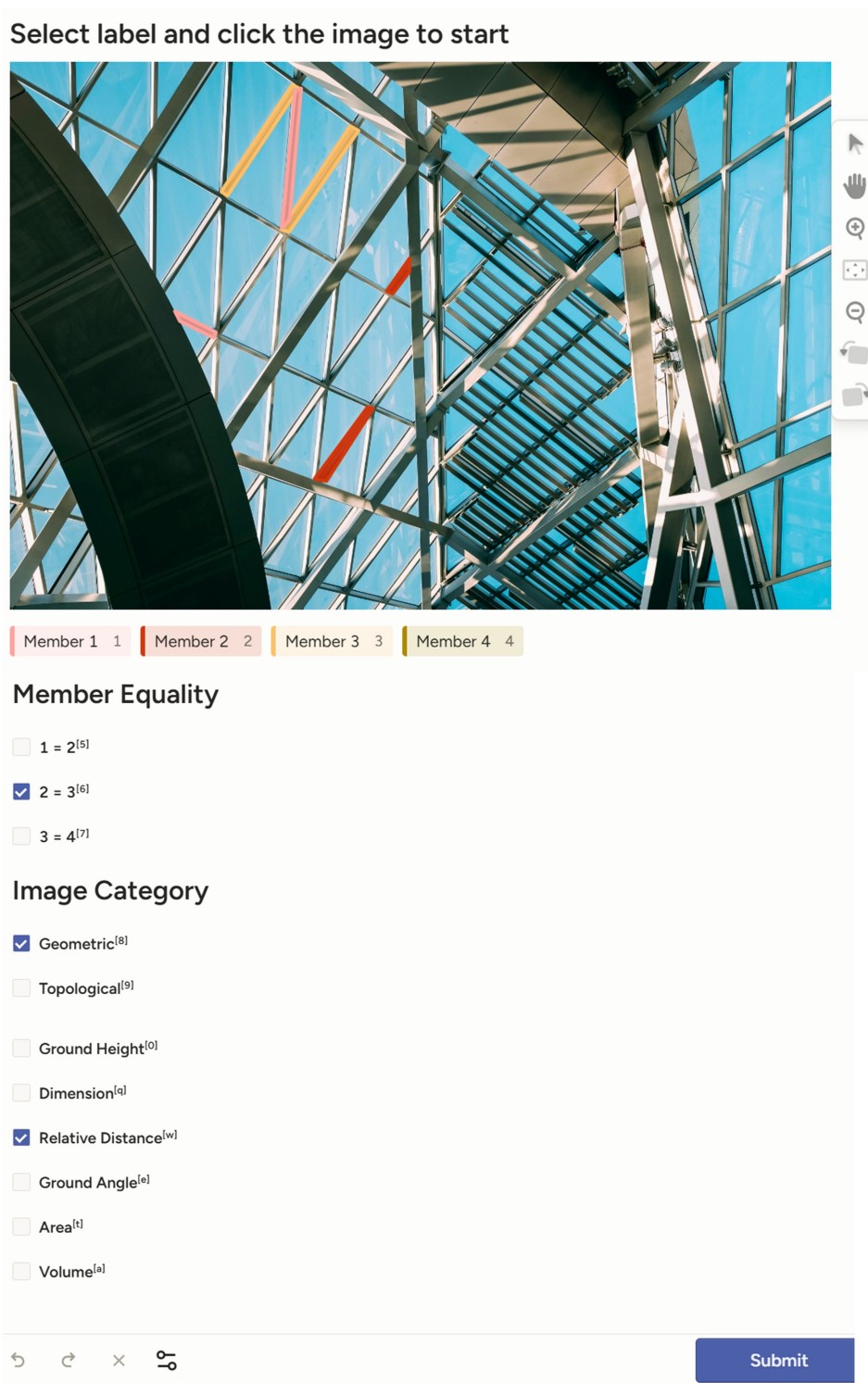

*Figure 7.* Screenshot of the data annotation interface, where annotators select members/groups, the task category, and specify equality relations.

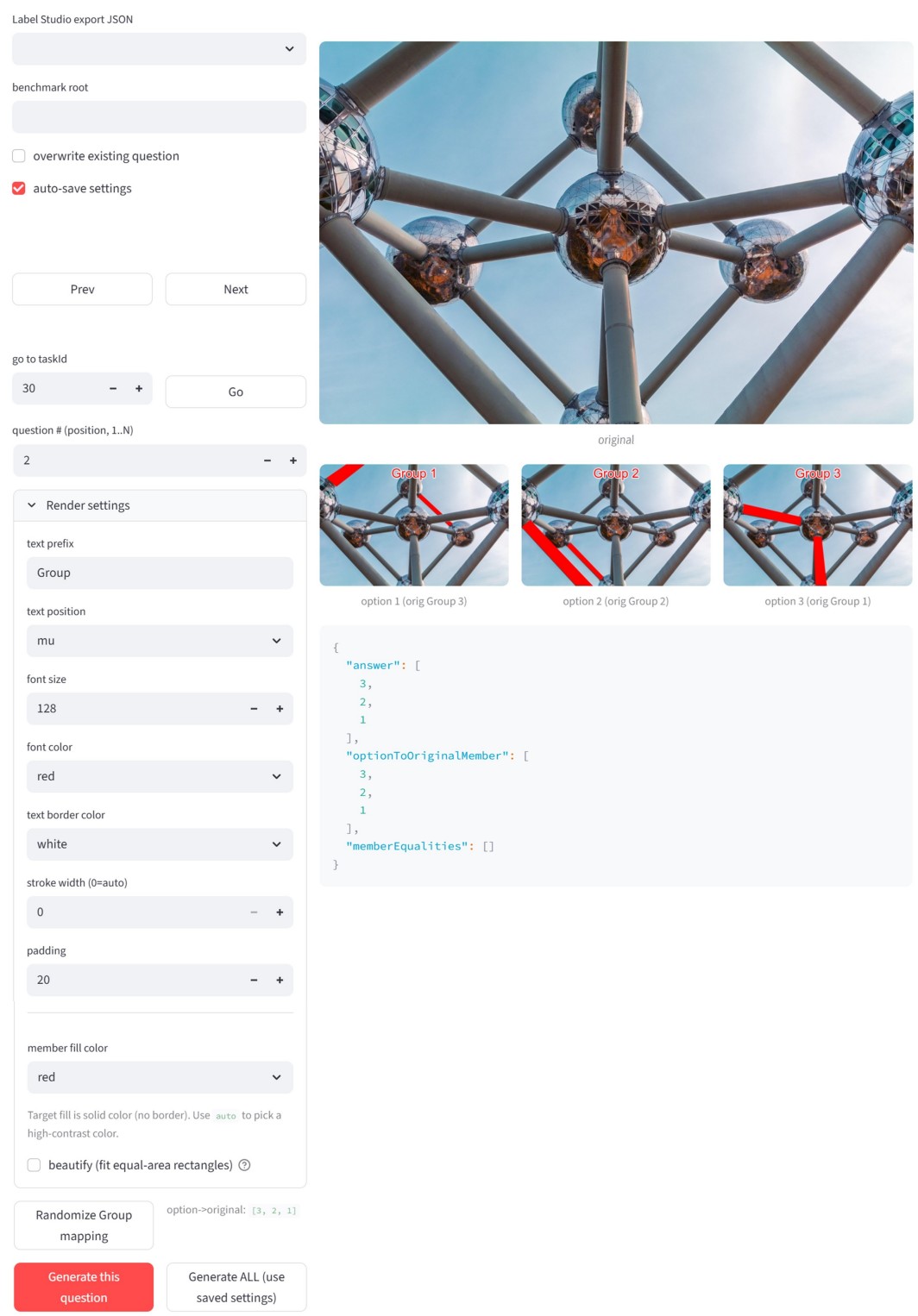

*Figure 8.* Screenshot of the manual review interface used during question generation, where annotators verify whether the annotated colors and text placements are appropriate.

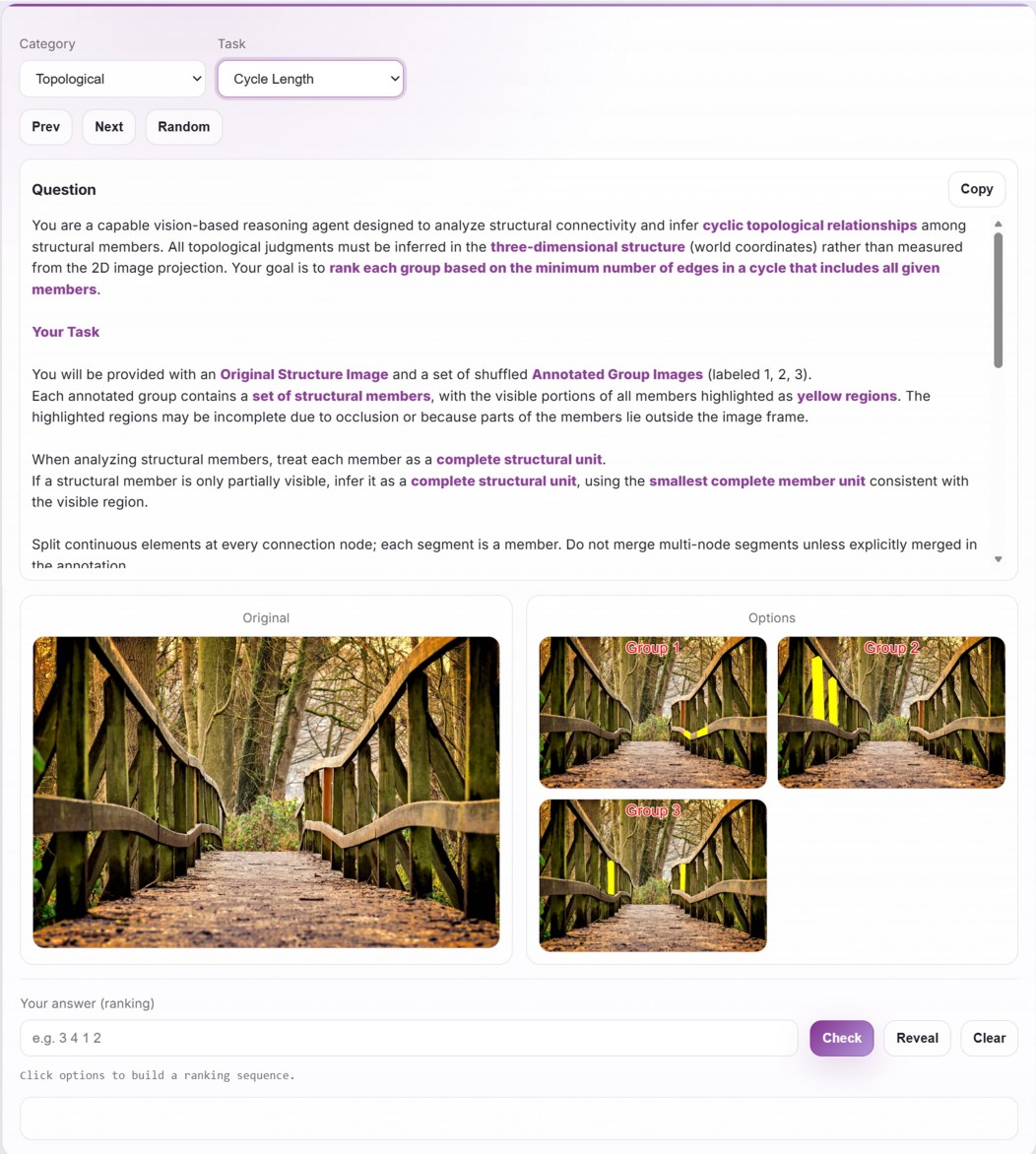

*Figure 9.* Screenshot of the independent review interface used for quality control, where annotators attempt the problem and provide their answers.

- **Style and rendering conventions.** Annotators use the default highlight colors in the interface; colors and label styling are standardized in a later post-processing step to ensure consistency across the dataset.

- **Task-specific criteria and special considerations.** Annotators must follow the task definitions and apply the following task-level notes to avoid systematic errors: (i) *Relative Distance* is defined along the straight line of the member (i.e., the member's axis), therefore curved components should not be selected as candidates; (ii) *Ground Angle* measures the angle relative to the ground plane, where members parallel to the ground have the smallest angle and members perpendicular to the ground have the largest; (iii) *Dimension* refers to the length along the member's principal direction (i.e., its main axis); (iv) *Cycle Length* is the number of edges in the smallest cycle that contains both members; (v) *Hop Distance* should not be too small; candidates are chosen to ensure non-trivial graph distances so that the task remains challenging; (vi) *Multi-View* instances require cross-view correspondence: one image labels the reference member as `0`, and the other image labels the target candidates as `1--3`. Annotators should first identify stable anchor regions visible in both views; if correspondence is unclear, the instance should be flagged for discussion.

### E.4. Data Format and Structure

To ensure consistency and facilitate reliable evaluation and analysis, `SSI-Bench` adopts a two-stage data organization: (i) per-instance storage for construction and inspection, and (ii) dataset-level exports for evaluation and release.

**Per-instance storage.**

- **JSON metadata.** Each instance is stored as a JSON object containing the question text, answer options, the ground-truth answer, task/category labels, difficulty, and relative file paths to the associated images.

- **Image files.** Images are stored separately and referenced by paths in the JSON entries. For each instance, we retain both (i) the *original* image(s) and (ii) the *annotated* image(s) rendered with highlights and text labels for evaluation. This design preserves raw visual content while enabling reproducible benchmarking with standardized visual cues.

**Dataset-level export and release.**

- **Evaluation-ready TSV.** To support automated benchmarking with VLMEvalKit (Duan et al., 2024), we aggregate all instances and export the benchmark into a TSV file tailored for evaluation. Each row corresponds to one instance and contains the minimal fields required for model inference and scoring, including the instance ID, question, answer label, category, and base64-encoded image content.

- **Hugging Face release format.** For convenient distribution and efficient loading, we additionally provide a packaged release on Hugging Face that stores the aggregated metadata and (binary-encoded) images in a Parquet-based format.

### E.5. Tie Handling and Evaluation Fairness

Some ranking questions in `SSI-Bench` may contain candidates that are equal or visually indistinguishable under the task-defined criterion. These ties are not treated as annotation ambiguities; rather, they are anticipated cases handled by a deterministic rule during both annotation and evaluation. Specifically, when two candidates are tied, the candidate with the smaller index is placed earlier in the ranking. This rule is consistently applied when constructing the ground-truth answer.

To ensure fairness, the same tie-breaking rule is explicitly stated in the prompts provided to VLMs. Therefore, models are not penalized for correctly recognizing equality; they are expected to follow the specified output convention once equality is identified. We use strict exact-match evaluation instead of a tie-aware metric because accepting both possible orders for tied candidates, e.g., accepting both $A < B$ and $B < A$ when $A = B$, would make it difficult to distinguish genuine recognition of equality from arbitrary guessing. The deterministic rule thus provides a clear and reproducible scoring protocol while preserving the ranking-based formulation of `SSI-Bench`.

### E.6. Semi-Automatic Expansion

Although `SSI-Bench` is constructed through a fully human-centered pipeline, its ranking-based design naturally supports semi-automatic expansion once candidate-level annotations are available. Instead of annotating each question from scratch, annotators can enrich an existing image with additional candidate members or groups, after which new ranking questions are

*Table 6.* Accuracy on the original and newly generated benchmark splits. The expanded set, generated from 100 images by adding two extra candidates per image (yielding 10 questions/image), produces performance comparisons highly consistent with the original benchmark.

| Split | Models | Geometric | | | | | | | Topological | | | Avg. | Rank |
|---|---|---|---|---|---|---|---|---|---|---|---|---|---|
| | | Grd. Ht. | Grd. Ang. | Dim. | Rel. Dist. | Area | Volume | M-View | Hop Dist. | Cyc. Len. | M-View | | |
| Original | Gemini-3-Flash | 37.14 | 38.61 | 35.00 | 41.75 | 27.00 | 25.77 | 33.96 | 32.98 | 34.69 | 28.13 | 33.60 | 1 |
| | GLM-4.6V | 9.52 | 21.78 | 16.00 | 30.10 | 28.00 | 21.65 | 26.42 | 25.53 | 20.41 | 22.92 | 22.20 | 2 |
| | InternVL3.5-241B-A28B | 3.81 | 6.93 | 6.00 | 28.16 | 25.00 | 16.49 | 21.70 | 28.72 | 23.47 | 23.96 | 18.30 | 3 |
| New | Gemini-3-Flash | 43.00 | 45.00 | 39.00 | 39.00 | 30.00 | 29.00 | 36.00 | 30.00 | 37.00 | 25.00 | 35.30 | 1 |
| | GLM-4.6V | 9.00 | 23.00 | 14.00 | 31.00 | 34.00 | 23.00 | 30.00 | 23.00 | 20.00 | 19.00 | 22.60 | 2 |
| | InternVL3.5-241B-A28B | 7.00 | 12.00 | 7.00 | 23.00 | 28.00 | 18.00 | 23.00 | 24.00 | 26.00 | 23.00 | 19.10 | 3 |

generated by automatically sampling candidate combinations under the same task definitions, prompt templates, tie-breaking rule, and rendering procedure as the original benchmark.

We conduct a pilot expansion to examine this scalability. Starting from 100 existing images, we add two extra candidates per image and automatically sample candidate combinations, yielding 10 questions per image and 1,000 new questions in total. The generated questions are then checked to remove ambiguous cases, unclear visual annotations, and overly trivial instances. As shown in Table 6, the newly generated split preserves the same model ranking as the original benchmark and yields highly consistent performance comparisons, suggesting that **SSI-Bench** can be scaled with substantially lower marginal annotation cost while maintaining its evaluation characteristics.

### E.7. Difficulty Annotation

To quantify instance difficulty in **SSI-Bench** in a scalable and objective manner, we use human *lead time* as the primary proxy. Lead time is defined as the elapsed time (in seconds) that an annotator stays on the annotation interface to design, verify, and finalize a question instance, including selecting candidates, confirming the ordering (and ties, if any), and completing the required metadata fields. Intuitively, instances that require longer lead time tend to involve more subtle geometric/topological judgments, heavier occlusion, or more careful cross-checking to ensure answer uniqueness.

**Robustness to idle time.** Raw lead time can be inflated by non-productive pauses (*e.g.*, brief interruptions while the interface remains open). To reduce the impact of such noise, we apply robust outlier filtering based on the Interquartile Range (IQR) rule, which is less sensitive to extreme values than mean/standard-deviation-based filtering.

**Outlier filtering.** Let $t$ denote the lead time of an annotation session. Over all collected sessions, we compute the first and third quartiles ($Q_1$ and $Q_3$) of $\{t\}$. In our data, $Q_1 = 101.46$ s and $Q_3 = 216.89$ s, yielding an interquartile range $\mathrm{IQR} = Q_3 - Q_1 = 115.43$ s. Following the standard 1.5×IQR heuristic, we define the upper outlier threshold as

$$t_{\max} = Q_3 + 1.5 \times \mathrm{IQR} = 390.03 \text{ s.} \tag{4}$$

Any session with $t > t_{\max}$ is treated as an outlier and excluded from subsequent difficulty labeling, since such cases are likely dominated by idle time rather than annotation complexity. (We do not apply a lower outlier bound because extremely short sessions are typically valid and correspond to straightforward instances.)

**Difficulty categorization.** After removing outliers, we assign a three-level difficulty label—*Easy*, *Medium*, and *Hard*—using a combination of (i) filtered lead time and (ii) empirical human performance during quality control. Figure 10 shows the post-filtering distribution.

- **Easy.** Instances with lead time in $[0, 150)$ seconds (515 instances). These questions are typically finalized quickly while still meeting our uniqueness and clarity constraints.

- **Medium.** Instances with lead time in $[150, 270)$ seconds (319 instances). These questions generally require additional verification, such as careful disambiguation under mild occlusion or more subtle comparisons.

- **Hard.** Instances that satisfy either of the following conditions (74 instances): (i) lead time in $[270, 400]$ seconds (bounded above by the outlier threshold), indicating sustained effort to ensure correctness and uniqueness; or (ii) the instance was answered incorrectly by at least one independent human reviewer during the quality control stage, suggesting that the instance is challenging even for humans under the same presentation format.

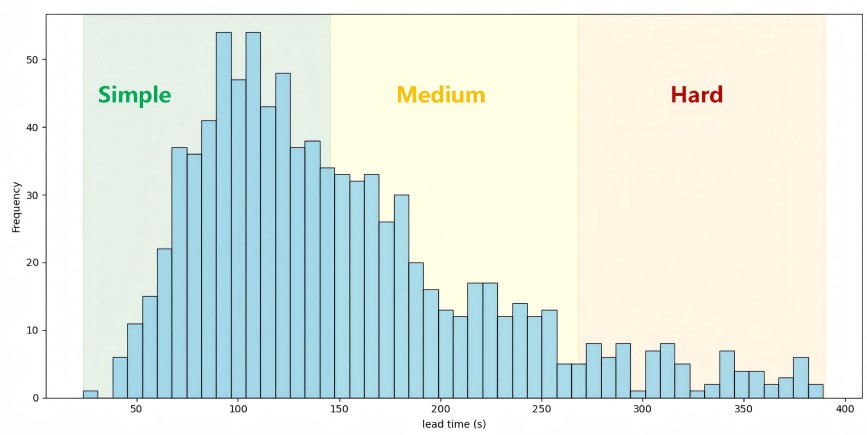

*Figure 10.* Distribution of human lead time for question annotation and the corresponding difficulty categorization.

## F. Additional Implementation Details and Experimental Results

### F.1. Evaluation Metrics

Each benchmark instance is a ranking question with $n$ candidate options (e.g., panels / sub-images / items) indexed by $\{0, 1, \ldots, n-1\}$. The ground-truth answer is a permutation $\pi^\star = [\pi_1^\star, \ldots, \pi_n^\star]$, where $\pi_k^\star$ denotes the index placed at rank $k$ (from best/most plausible to worst/least plausible, following the instruction of the corresponding QA type). The model prediction is a list $\hat{\pi} = [\hat{\pi}_1, \ldots, \hat{\pi}_n]$.

**Output parsing and validity.** Models are instructed to output a *parsable Python list* of integers. We parse the first valid list appearing in the output using a strict Python literal parser. A prediction is considered *valid* if and only if: (i) its length equals $n$; (ii) all entries are integers; (iii) the set of entries equals $\{0, 1, \ldots, n-1\}$ (i.e., it is a permutation with no duplicates or missing indices). If the output is not valid, we treat the instance as incorrect for all metrics (i.e., score 0 for Taskwise Accuracy and Pairwise Accuracy). This choice avoids introducing heuristic post-processing that may unfairly benefit certain models.

**Taskwise Accuracy.** Taskwise Accuracy measures exact match on the full permutation:

$$\text{Acc}_{\text{task}} \;=\; \frac{1}{N} \sum_{i=1}^{N} \mathbf{1}\big[\hat{\pi}^{(i)} = \pi^{\star(i)}\big], \tag{5}$$

where $N$ is the number of instances, and $\mathbf{1}[\cdot]$ is the indicator function. This metric is strict: any deviation at any position leads to 0 for the instance.

**Pairwise Accuracy.** Taskwise exact match can be overly harsh when a prediction is close to the correct ordering. We therefore report Pairwise Accuracy, which evaluates *pairwise ordering consistency* between the predicted ranking and the ground truth. For any two distinct items $(a, b)$, define the induced order by a permutation $\pi$ as:

$$a \prec_\pi b \;\;\Leftrightarrow\;\; \text{pos}_\pi(a) < \text{pos}_\pi(b), \tag{6}$$

where $\text{pos}_\pi(x)$ returns the rank position of item $x$ in permutation $\pi$. Pairwise Accuracy for an instance is the fraction of item pairs whose relative order matches the ground truth:

$$\text{Acc}_{\text{pair}}(\hat{\pi}, \pi^\star) \;=\; \frac{1}{\binom{n}{2}} \sum_{0 \le a < b \le n-1} \mathbf{1}\Big[(a \prec_{\hat{\pi}} b \;\wedge\; a \prec_{\pi^\star} b) \;\vee\; (b \prec_{\hat{\pi}} a \;\wedge\; b \prec_{\pi^\star} a)\Big]. \tag{7}$$

We report the dataset-level Pairwise Accuracy as the average over instances:

$$\text{Acc}_{\text{pair}} \;=\; \frac{1}{N} \sum_{i=1}^{N} \text{Acc}_{\text{pair}}\Big(\hat{\pi}^{(i)}, \pi^{\star(i)}\Big). \tag{8}$$

*Table 7.* Human performance comparison on high- vs. low-resolution images. For each task category, 10 questions are randomly sampled, resulting in 100 questions in total.

| Resolution | Geometric | | | | | | | Topological | | | Avg. |
|---|---|---|---|---|---|---|---|---|---|---|---|
| | Grd. Ht. | Grd. Ang. | Dim. | Rel. Dist. | Area | Volume | M-View | Hop Dist. | Cyc. Len. | M-View | |
| 1920px | 90 | 90 | 100 | 90 | 90 | 100 | 80 | 100 | 90 | 90 | 92 |
| 512px | 100 | 90 | 90 | 90 | 80 | 100 | 80 | 90 | 90 | 90 | 90 |

**Random baseline.** We report the expected performance of a uniformly random ranking over $n$ items.

**(i) Taskwise Accuracy.** Since there are $n!$ possible permutations, the probability of exactly matching the ground-truth permutation is:

$$\mathbb{E}\left[\text{Acc}_{\text{task}}^{\text{rand}}\right] = \frac{1}{n!}. \tag{9}$$

**(ii) Pairwise Accuracy.** For any pair $(a, b)$, a uniformly random permutation puts $a$ before $b$ with probability $1/2$. Therefore, the expected fraction of correctly ordered pairs is:

$$\mathbb{E}\left[\text{Acc}_{\text{pair}}^{\text{rand}}\right] = \frac{1}{2}. \tag{10}$$

**Reporting.** All metrics are computed with temperature 0 and the same prompt template per QA type, as described in Section 4.1. We report both Taskwise Accuracy and Pairwise Accuracy to capture strict end-to-end correctness and softer ordering quality, respectively.

### F.2. Benchmark Models

We conduct a comprehensive evaluation across 31 VLMs, covering both proprietary and open-source families, broad parameter scales, and recent architectural advances. For proprietary systems, we evaluate Google DeepMind's Gemini series (Gemini-3-Pro and Gemini-3-Flash; Gemini-2.5-Pro and Gemini-2.5-Flash) (Google DeepMind, 2025b;a), OpenAI's GPT family (GPT-5.2, GPT-5 mini, GPT-4.1, and GPT-4o) (OpenAI, 2025c;b;a; 2024), Anthropic's Claude-Sonnet-4.5 (Anthropic, 2025), and ByteDance Seed-1.8 (ByteDance Seed, 2025). On the open-source side, we systematically assess 21 state-of-the-art models spanning multiple families and sizes: the GLM-V series (GLM-4.6V, GLM-4.6V-Flash, GLM-4.5V) (Hong et al., 2026), Qwen3-VL across MoE and dense variants (Qwen3-VL-235B-A22B, 30B-A3B, 8B/4B/2B) (Bai et al., 2025), InternVL3.5 across a wide range of scales (InternVL3.5-241B-A28B, 30B-A3B, 38B/14B/8B/4B/2B) (Wang et al., 2025b), Meta's Llama-4-Scout-17B-16E (Meta, 2025) (Meta AI, 2025), Google's Gemma-3 family (27B/12B/4B) (Kamath et al., 2025), and LLaVA-OneVision in 72B and 7B settings (Li et al., 2024).

### F.3. Implementation Details for Model Evaluation

**Execution setup.** Proprietary models are accessed via their official APIs to ensure standardized inference behavior and fair comparison across providers. In contrast, open-source models are deployed locally on our in-house inference stack on a single node equipped with $8\times$ NVIDIA A100 80GB GPUs.

**Image preprocessing.** To ensure a consistent input budget across models, all images are resized such that the longer side is at most 512 pixels while preserving the aspect ratio. We apply the same preprocessing pipeline for all evaluated VLMs. This resolution is chosen to control inference cost and maintain a comparable visual-token budget across models. To verify that it does not remove essential visual information for human spatial reasoning, we conduct an additional human study on 100 randomly sampled questions, comparing the original high-resolution images with their 512px resized versions. As shown in Table 7, human accuracy only decreases slightly from 92% to 90%, indicating that the 512px preprocessing preserves sufficient detail for solving **SSI-Bench** questions.

**Evaluation framework.** We implement the evaluation pipeline using VLMEVALKIT, which standardizes prompt construction, model invocation, output parsing, and metric computation across different backends (API-based and locally deployed).

*Table 8.* Evaluation on `SSI-Bench` (Pairwise Accuracy). We highlight the best and second best results within each category (Proprietary or Open-source Models). Abbreviations: Grd. Ht. = Ground Height; Grd. Ang. = Ground Angle; Dim. = Dimension; Rel. Dist. = Relative Distance; Area = Area; Volume = Volume; M-View = Multi-View; Hop Dist. = Hop Distance; Cyc. Len = Cycle Length.

| MODELS | GEOMETRIC | | | | | | | TOPOLOGICAL | | | AVG. |
|---|---|---|---|---|---|---|---|---|---|---|---|
| | Grd. Ht. | Grd. Ang. | Dim. | Rel. Dist. | Area | Volume | M-View | Hop Dist. | Cyc. Len. | M-View | |
| *Proprietary Models* | | | | | | | | | | | |
| GEMINI-3-PRO | 69.52 | 78.38 | 74.00 | 66.99 | 56.67 | 59.79 | 61.64 | 61.35 | 62.93 | 59.38 | 65.15 |
| GEMINI-3-FLASH | 76.03 | 78.22 | 74.50 | 68.93 | 56.33 | 58.08 | 62.26 | 64.54 | 66.67 | 60.76 | 66.73 |
| GEMINI-2.5-PRO | 66.98 | 76.24 | 64.17 | 65.70 | 50.33 | 55.67 | 62.26 | 56.74 | 62.24 | 58.33 | 61.98 |
| GEMINI-2.5-FLASH | 67.30 | 70.13 | 66.00 | 55.02 | 52.67 | 46.39 | 59.43 | 54.96 | 58.84 | 58.68 | 59.05 |
| GPT-5.2 | 72.86 | 78.88 | 77.67 | 72.17 | 56.00 | 51.20 | 63.52 | 61.35 | 65.99 | 57.29 | 65.85 |
| GPT-5 MINI | 69.68 | 74.09 | 67.67 | 71.52 | 58.67 | 49.83 | 55.03 | 53.55 | 57.82 | 58.68 | 61.80 |
| GPT-4.1 | 63.33 | 59.74 | 68.17 | 52.43 | 56.67 | 50.52 | 49.37 | 58.51 | 56.80 | 58.33 | 57.37 |
| GPT-4O | 66.67 | 63.70 | 65.33 | 57.93 | 54.00 | 52.92 | 50.00 | 57.45 | 54.42 | 60.76 | 58.33 |
| CLAUDE-SONNET-4.5 | 53.49 | 59.90 | 64.17 | 61.49 | 49.67 | 47.42 | 53.46 | 53.19 | 55.78 | 57.29 | 55.62 |
| SEED-1.8 | 67.46 | 74.09 | 69.33 | 69.58 | 48.67 | 50.86 | 57.55 | 60.99 | 67.69 | 61.11 | 62.80 |
| *Open-source Models* | | | | | | | | | | | |
| GLM-4.6V | 63.65 | 69.47 | 64.83 | 62.78 | 60.33 | 50.86 | 58.18 | 58.16 | 55.10 | 55.90 | 60.02 |
| GLM-4.6V-FLASH | 64.92 | 65.02 | 62.00 | 59.55 | 59.00 | 51.55 | 51.89 | 59.22 | 54.42 | 60.42 | 58.82 |
| GLM-4.5V | 66.67 | 73.60 | 67.33 | 59.87 | 59.67 | 49.83 | 53.46 | 58.16 | 57.14 | 57.99 | 60.43 |
| QWEN3-VL-235B-A22B | 62.22 | 67.49 | 67.83 | 59.87 | 55.33 | 46.05 | 54.72 | 62.41 | 53.40 | 57.99 | 58.77 |
| QWEN3-VL-30B-A3B | 54.60 | 53.47 | 64.17 | 58.58 | 61.33 | 50.86 | 54.09 | 59.57 | 60.54 | 59.38 | 57.62 |
| QWEN3-VL-8B | 54.29 | 50.83 | 61.00 | 58.25 | 56.00 | 51.55 | 57.55 | 56.38 | 60.54 | 58.33 | 56.47 |
| QWEN3-VL-4B | 55.56 | 52.31 | 59.33 | 58.58 | 57.67 | 50.52 | 55.97 | 59.57 | 61.56 | 57.99 | 56.88 |
| QWEN3-VL-2B | 53.17 | 52.81 | 59.17 | 59.22 | 63.33 | 50.52 | 44.65 | 61.70 | 60.54 | 26.04 | 53.13 |
| INTERNVL3.5-241B-A28B | 50.63 | 52.31 | 60.00 | 59.55 | 59.00 | 48.45 | 52.52 | 60.28 | 54.76 | 52.08 | 54.93 |
| INTERNVL3.5-30B-A3B | 54.60 | 52.64 | 60.00 | 58.90 | 63.33 | 50.17 | 38.68 | 60.28 | 60.54 | 58.68 | 55.65 |
| INTERNVL3.5-38B | 55.56 | 52.15 | 66.00 | 56.63 | 63.33 | 50.17 | 55.66 | 59.93 | 55.44 | 58.33 | 57.30 |
| INTERNVL3.5-14B | 53.33 | 50.83 | 59.50 | 50.81 | 60.33 | 50.86 | 56.60 | 53.90 | 55.44 | 57.99 | 54.95 |
| INTERNVL3.5-8B | 50.95 | 52.15 | 54.67 | 53.72 | 63.67 | 50.52 | 58.18 | 58.16 | 60.54 | 57.99 | 56.02 |
| INTERNVL3.5-4B | 54.29 | 51.49 | 55.67 | 54.37 | 61.00 | 50.17 | 30.82 | 52.84 | 59.18 | 50.35 | 51.90 |
| INTERNVL3.5-2B | 50.63 | 46.53 | 55.50 | 49.84 | 58.67 | 42.96 | 16.35 | 49.65 | 34.69 | 1.74 | 40.70 |
| LLAMA-4-SCOUT-17B-16E | 56.98 | 68.48 | 64.50 | 57.61 | 62.67 | 47.08 | 56.60 | 56.74 | 59.86 | 59.03 | 58.98 |
| GEMMA-3-27B | 51.75 | 51.98 | 57.33 | 58.25 | 63.33 | 50.17 | 55.97 | 59.57 | 61.22 | 58.33 | 56.75 |
| GEMMA-3-12B | 46.83 | 50.00 | 47.67 | 47.90 | 58.67 | 48.80 | 56.60 | 57.45 | 61.22 | 59.03 | 53.33 |
| GEMMA-3-4B | 54.92 | 52.64 | 58.50 | 56.31 | 63.67 | 50.52 | 56.92 | 59.93 | 54.42 | 56.60 | 56.43 |
| LLAVA-ONEVISION-72B | 49.68 | 51.98 | 50.50 | 50.81 | 62.00 | 49.48 | 56.92 | 54.26 | 55.10 | 59.38 | 53.98 |
| LLAVA-ONEVISION-7B | 52.22 | 50.66 | 45.83 | 49.84 | 58.00 | 48.45 | 57.23 | 58.51 | 54.76 | 56.25 | 53.15 |
| *Baselines* | | | | | | | | | | | |
| RANDOM GUESSING | 50.00 | 50.00 | 50.00 | 50.00 | 50.00 | 50.00 | 50.00 | 50.00 | 50.00 | 50.00 | 50.00 |
| **HUMAN PERFORMANCE** | **99.05** | **98.18** | **97.83** | **97.09** | **99.00** | **95.88** | **96.23** | **97.16** | **95.58** | **90.97** | **96.73** |

## F.4. Human Evaluation Setup

Human performance is measured as the average accuracy of six adult participants who were not involved in the data annotation process. The six participants collectively completed all 1,000 benchmark questions. During evaluation, participants are presented with the question and the corresponding images simultaneously (including the original image and all annotated variants, identical to the visual inputs provided to the VLMs). They are given unlimited time to answer each question to the best of their ability and may revisit the images as many times as needed. To facilitate answer submission, participants provide their rankings by clicking the options in order to form a permutation.

## F.5. Generality Analysis

Although `SSI-Bench` focuses on structure-centric spatial reasoning, it is not intended to evaluate specialized engineering expertise. We examine this from two perspectives: human performance across different backgrounds and model-performance consistency with general spatial benchmarks.

*Table 9.* Performance of human evaluators on `SSI-Bench`. The non-expert group received a brief introduction under 10 minutes to structural common sense prior to testing. For each task category, 10 questions were randomly sampled, resulting in 100 questions in total.

| Participant Group | Geometric | | | | | | | Topological | | | Avg. |
|---|---|---|---|---|---|---|---|---|---|---|---|
| | Grd. Ht. | Grd. Ang. | Dim. | Rel. Dist. | Area | Volume | M-View | Hop Dist. | Cyc. Len. | M-View | |
| Basic Eng. Background | 90 | 90 | 100 | 90 | 90 | 100 | 80 | 100 | 90 | 90 | 92 |
| Non-Expert (Layperson) | 90 | 80 | 100 | 90 | 80 | 90 | 90 | 100 | 100 | 90 | 91 |

*Table 10.* Performance of various models on `SSI-Bench` and other general spatial benchmarks. The models are sorted from left to right in descending order based on their performance on `SSI-Bench`. "–" indicates that the model's performance on the corresponding benchmark was not reported.

| Benchmark | Gemini-3-Pro | Gemini-2.5-Pro | GPT-4o | InternVL3.5-8B | LLaVA-Onevision-7B |
|---|---|---|---|---|---|
| SSI-Bench | 29.50 | 26.10 | 22.60 | 20.20 | 16.50 |
| DSI-Bench | – | 46.90 | 37.23 | 36.41 | – |
| MMSI-Video-Bench | 38.00 | – | 31.60 | – | – |
| MMSI-Bench | – | 36.90 | 30.30 | – | 24.50 |
| ViewSpatial-Bench | – | – | 34.98 | – | 27.49 |
| CVBench | – | – | 69.10 | – | 52.60 |
| VSI-Bench | – | – | 34.00 | – | 32.40 |

**Non-expert human performance.** To examine whether `SSI-Bench` requires specialized engineering expertise, we further conduct a pilot study with non-expert participants who do not have structural-engineering backgrounds. Before testing, they receive only a brief introduction to basic structural common sense, lasting less than 10 minutes. We randomly sample 10 questions from each task category, resulting in 100 questions in total. As shown in Table 9, non-expert participants achieve 91% accuracy, which is close to the 92% accuracy of participants with basic engineering background on the same subset. This suggests that `SSI-Bench` primarily evaluates spatial reasoning rather than domain-specific engineering knowledge. In particular, Area and Volume questions do not require advanced engineering computation. They are designed so that humans can solve them through intuitive 3D comparison after identifying the relevant structural components.

**Consistency with general spatial benchmarks.** We further compare model performance on `SSI-Bench` with reported results on representative spatial benchmarks that cover broader scene-centric or video-based settings. As shown in Table 10, models with stronger performance on `SSI-Bench` generally also perform better on these existing benchmarks when overlapping results are available. This trend suggests that `SSI-Bench` captures a relevant dimension of general spatial intelligence, while still testing a complementary structure-centric capability that is underrepresented in prior evaluations. Because public reports cover different model subsets across benchmarks, we treat this comparison as a consistency analysis rather than a definitive correlation estimate.

### F.6. Pairwise Accuracy Results

We additionally report **Pairwise Accuracy** results on `SSI-Bench` in Table 8. Pairwise Accuracy evaluates models in a two-choice, comparative setting using the same geometric and topological attributes as in the main table. We retain the same model grouping (Proprietary vs. Open-source) and highlight the best and second-best performance within each group to enable fair, within-group comparisons.

Overall, **Pairwise Accuracy and Taskwise Accuracy lead to highly consistent conclusions about model capability**. Within the proprietary group, models that perform strongly under Taskwise Accuracy remain strong under Pairwise Accuracy (e.g., Gemini-3-Flash continues to achieve the best average performance), while weaker models remain near the bottom (e.g., Claude-Sonnet-4.5). This consistency suggests that the two evaluation protocols largely agree on global model rankings, and that Pairwise Accuracy provides a reliable complementary perspective on the same underlying skills.

At the same time, we observe **localized discrepancies**, primarily among mid-tier models. A key reason is that Taskwise Accuracy typically requires a model to resolve all relevant comparisons within a problem correctly (e.g., producing a globally consistent ordering across multiple pairs), whereas Pairwise Accuracy scores each comparison independently. As a result, some mid-tier models can achieve reasonably high Pairwise Accuracy by getting many individual pairwise judgments right, yet still underperform on Taskwise Accuracy because they fail to handle specific "trap" cases or corner conditions embedded in the full problem (e.g., subtle ambiguities, distractors, or exceptions that break global consistency). In this sense, Taskwise Accuracy places greater emphasis on robust pitfall recognition and end-to-end consistency, while Pairwise

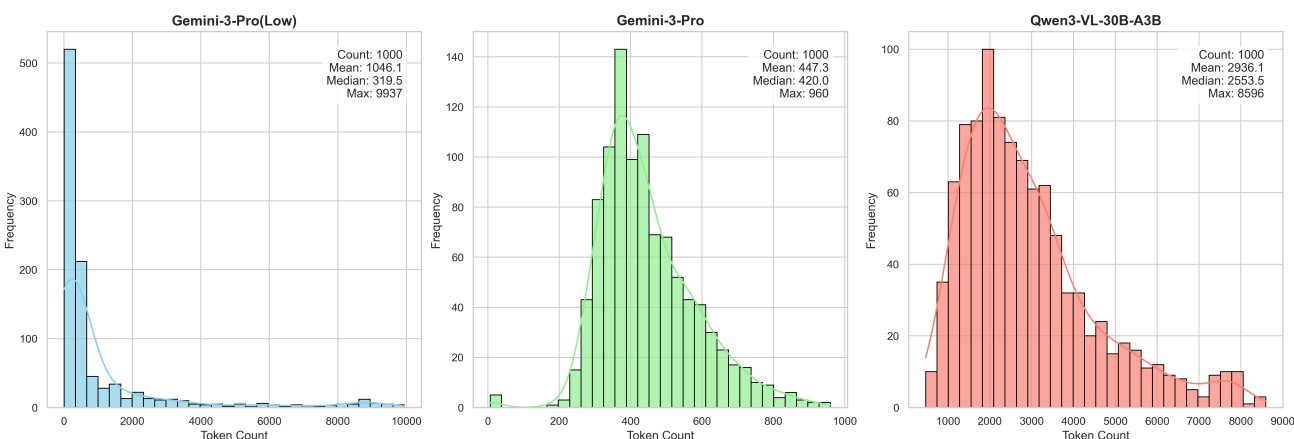

*Figure 11.* Distributions of the number of tokens used for explicit thinking for different models.

*Table 11.* Complete evaluation results of thinking on **SSI-Bench** (Task Accuracy). Abbreviations: Grd. Ht. = Ground Height; Grd. Ang. = Ground Angle; Dim. = Dimension; Rel. Dist. = Relative Distance; Area = Area; Volume = Volume; M-View = Multi-View; Hop Dist. = Hop Distance; Cyc. Len = Cycle Length.

| MODELS | GEOMETRIC | | | | | | | TOPOLOGICAL | | | AVG. |
|---|---|---|---|---|---|---|---|---|---|---|---|
| | Grd. Ht. | Grd. Ang. | Dim. | Rel. Dist. | Area | Volume | M-View | Hop Dist. | Cyc. Len. | M-View | |
| GEMINI-3-PRO (HIGH) | 25.71 | 37.62 | 28.00 | 33.01 | 24.00 | 27.84 | 31.13 | 35.11 | 30.61 | 21.88 | 29.50 |
| GEMINI-3-PRO (LOW) | 22.86 | 30.69 | 27.00 | 32.04 | 19.00 | 30.93 | 32.08 | 21.28 | 27.55 | 27.08 | 27.10 |
| QWEN3-VL-30B-A3B-THINKING | 5.71 | 12.87 | 14.00 | 33.01 | 28.00 | 21.65 | 21.70 | 27.66 | 30.61 | 30.21 | 22.50 |
| QWEN3-VL-30B-A3B-INSTRUCT | 5.71 | 7.92 | 12.00 | 28.16 | 29.00 | 17.53 | 22.64 | 28.72 | 28.57 | 27.08 | 20.60 |

Accuracy more directly reflects local comparative reasoning. Taken together, the results suggest that the two metrics are broadly aligned yet complementary: Taskwise Accuracy is stricter due to its holistic correctness requirement, whereas Pairwise Accuracy better isolates comparison-based geometric/topological intuition.

### F.7. Complete Evaluation Results of Thinking

As shown in Figure 11, different models exhibit markedly different distributions in the number of tokens used for explicit thinking, indicating diverse reasoning budgets and generation behaviors.

Table 11 reports the complete task accuracy on **SSI-Bench** for a set of representative thinking-enabled and instruct-style variants. We organize the results into two groups and highlight the best and second-best scores *within each group* for every sub-task and the overall average. For Gemini-3-Pro, the *high* setting yields higher accuracy on most geometric and topological dimensions, while the *low* setting remains competitive and even performs better on *Volume*, *Geometric M-View*, and *Topological M-View*. For Qwen3-VL-30B-A3B, the *Thinking* variant provides consistent gains over the *Instruct* variant on the majority of sub-tasks as well as the overall average, whereas *Instruct* shows advantages on *Area*, *Geometric M-View*, and *Hop Dist.*.

## G. Representative **SSI-Bench** Samples from Each Category

For clarity and readability, the questions and rationales shown in the main text are simplified; this section provides the complete versions for each category. We include one full example for each of the 11 sub-categories: Ground Height, Ground Angle, Dimension, Relative Distance, Area, Volume, Multi-View (Geo.), Hop Distance, Cycle Length, and Multi-View (Topo.). Note that Multi-View (Geo.) is the multi-view variant of Relative Distance. In contrast, Multi-View (Topo.) comprises two prompt types, corresponding to the multi-view variants of Hop Distance and Cycle Length, respectively. These representative samples are illustrated in Figure 12–Figure 22.

# H. Case Studies

In this section, we present additional complete reasoning processes of current VLMs to illustrate their spatial reasoning deficiencies more intuitively. As shown in Figure 23, Figure 24, and Figure 25, we highlight four types of errors that identify dominant model bottlenecks: Member-Extent Errors(orange), Object-Recognition Errors(red), Computational and Comparison-Logic Errors(green), and 3D Spatial-Logic Errors(blue).

# I. Limitations

To ensure high data quality, clear supervision, and reduced shortcut cues, we adopted a fully human-centered pipeline to curate images and manually annotate question–answer pairs (including rankings and ties). This design choice inevitably limits scalability: constructing and verifying instances requires substantial expert time and careful quality control, making it less scalable than fully automated generation pipelines. Nevertheless, we believe the current benchmark scale (1,000 samples) is sufficient for meaningful assessment at the present stage, since today's VLMs still exhibit a large gap to human performance on **SSI-Bench** and do not appear close to saturation. As models improve and performance begins to saturate, expanding the benchmark—potentially with more diverse structures and harder cases—may become necessary, but reaching that point will likely require either further research breakthroughs or additional human annotation effort. Importantly, the metadata annotation and question generation stages also support semi-automatic expansion: once richer candidate annotations are available for an image, new ranking questions can be generated by sampling candidate combinations and then verified by humans, as discussed in Appendix E.6. This reduces the marginal cost of scaling, although human review remains necessary to preserve answer uniqueness, visual clarity, and resistance to superficial shortcut cues. For test-time evaluation, we prioritize data quality and reliability, which is why we opted for manual curation over fully automated pipelines.

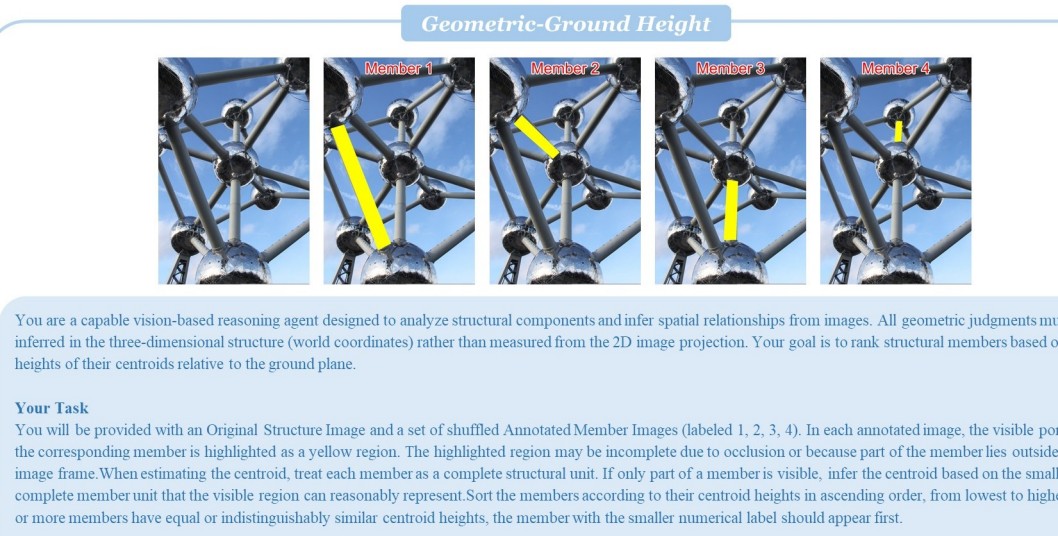

*Figure 12.* Full example for geometric-ground height category.

### Geometric-Ground Angle

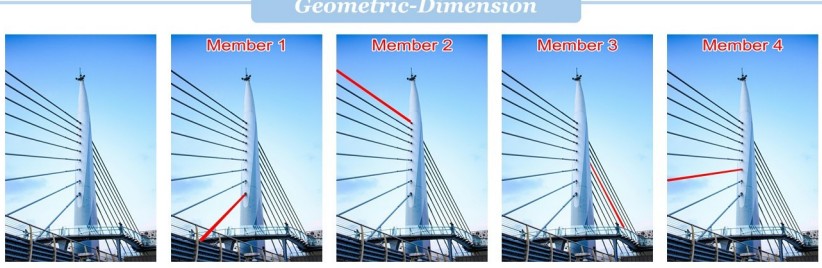

You are a capable vision-based reasoning agent designed to analyze structural components and infer geometric relationships from images. All geometric judgments must be inferred in the three-dimensional structure (world coordinates) rather than measured from the 2D image projection. Your goal is to rank structural members according to the angles between their main directions and the ground plane.

**Your Task**
You will be provided with an Original Structure Image and a set of shuffled Annotated Member Images (labeled 1, 2, 3, 4). In each annotated image, the visible portion of the corresponding member is highlighted as a yellow region. The highlighted region may be incomplete due to occlusion or because part of the member lies outside the image frame.When estimating the angle, treat each member as a complete structural unit. If only part of a member is visible, infer the main direction based on the smallest complete member unit that the visible region can reasonably represent.Sort the members according to the angle between their main direction and the ground plane in ascending order, from smallest angle to largest angle. A member parallel to the ground plane has the smallest angle, while a member perpendicular to the ground plane has the largest angle.If two or more members have equal or indistinguishably similar angles, the member with the smaller numerical label should appear first.

**Output Format**
Your response must be only a Python list of integers representing the order of member labels from smallest angle to largest angle. Do not include any other text, reasoning, or explanation.**Example:** If you determine the order is Member 2 (smallest angle), Member 1, Member 4, Member 3 (largest angle), your output must be:[2, 1, 4, 3]

Now please provide your answer in the requested format.

**A:** [3, 2, 4, 1]

*Figure 13.* Full example for geometric-ground angle category.

### Geometric-Dimension

You are a capable vision-based reasoning agent designed to analyze structural components and infer geometric properties from images. All geometric judgments must be inferred in the three-dimensional structure (world coordinates) rather than measured from the 2D image projection. Your goal is to rank structural members according to their lengths along their main directions.

**Your Task**
You will be provided with an Original Structure Image and a set of shuffled Annotated Member Images (labeled 1, 2, 3, 4). In each annotated image, the visible portion of the corresponding member is highlighted as a red region. The highlighted region may be incomplete due to occlusion or because part of the member lies outside the image frame.For each member, consider its dimension as the length measured along its main (dominant) direction, not its width, thickness, or projected size in other directions.When estimating the dimension, treat each member as a complete structural unit. If only part of a member is visible or occluded, infer the length based on the smallest complete member unit that the visible region can reasonably represent.Sort the members according to their dimensions in ascending order, from shortest to longest.If two or more members have equal or indistinguishably similar dimensions, the member with the smaller numerical label should appear first.

**Output Format**
Your response must be only a Python list of integers representing the order of member labels from shortest to longest. Do not include any other text, reasoning, or explanation.**Example:** If you determine the order is Member 4 (shortest), Member 1, Member 3, Member 2 (longest), your output must be:[4, 1, 3, 2]

Now please provide your answer in the requested format.

**A:** [1, 4, 3, 2]

*Figure 14.* Full example for geometric-dimension category.

### Geometric-Relative Distance

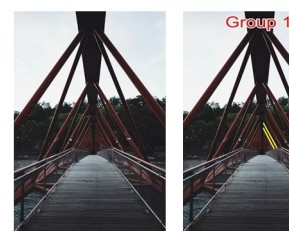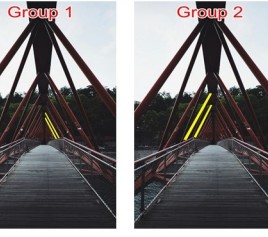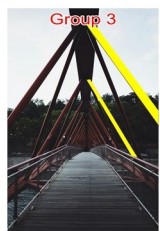

You are a capable vision-based reasoning agent designed to analyze structural components and infer spatial relationships from images. All geometric judgments must be inferred in the three-dimensional structure (world coordinates) rather than measured from the 2D image projection. Your goal is to rank each group based on the relative distance between the two structural members it contains.

**Your Task**
You will be provided with an Original Structure Image and a set of shuffled Annotated Group Images (labeled 1, 2, 3). Each annotated group contains two structural members, with the visible portions of both members highlighted as yellow regions. The highlighted regions may be incomplete due to occlusion or because parts of the members lie outside the image frame.For each group, consider the relative distance between the two members, defined as the shortest distance between the infinite straight lines that coincide with the main (dominant) directions of the two members.If the two lines intersect, their relative distance is defined as 0.Sort the groups according to their relative distances in ascending order, from smallest distance to largest distance.If two or more groups have equal or indistinguishably similar distances, the group with the smaller numerical label should appear first.

**Output Format**
Your response must be only a Python list of integers representing the order of group labels from smallest distance to largest distance. Do not include any other text, reasoning, or explanation.**Example:** If you determine the order is Group 3 (smallest distance), Group 1, Group 2 (largest distance), your output must be:[3, 1, 2]

Now please provide your answer in the requested format.

**A:** [1, 2, 3]

*Figure 15.* Full example for geometric-relative distance category.

### Geometric-Area

You are a capable vision-based reasoning agent designed to analyze structural components and infer geometric properties from images. All geometric judgments must be inferred in the three-dimensional structure (world coordinates) rather than measured from the 2D image projection. Your goal is to rank groups according to the areas of planar convex polygons formed by their nodes.

**Your Task**
You will be provided with an Original Structure Image and a set of shuffled Annotated Group Images (labeled 1, 2, 3). Each annotated group contains a set of nodes, highlighted as cyan points, which together define a planar convex polygon. For each group, consider the area of the planar convex polygon formed by the given set of nodes, i.e., the area of the convex hull of the nodes. Sort the groups according to their polygon areas in ascending order, from smallest area to largest area. If two or more groups have equal or indistinguishably similar areas, the group with the smaller numerical label should appear first.

**Output Format**
Your response must be only a Python list of integers representing the order of group labels from smallest area to largest area. Do not include any other text, reasoning, or explanation. **Example:** If you determine the order is Group 2 (smallest area), Group 1, Group 3 (largest area), your output must be: [2, 1, 3]

Now please provide your answer in the requested format.

**A:** [3, 1, 2]

*Figure 16.* Full example for geometric-area category.

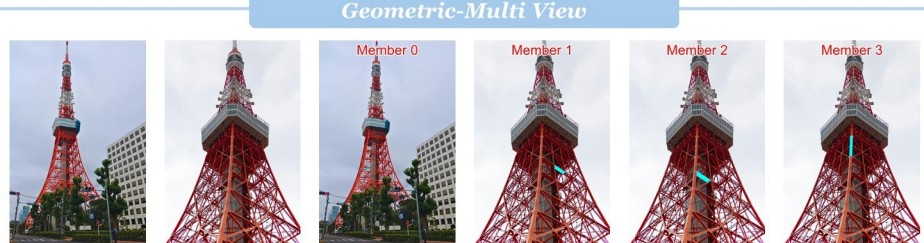

*Figure 17.* Full example for geometric-volume category.

*Figure 18.* Full example for geometric-multi view category.

**Topological - Hop Distance**

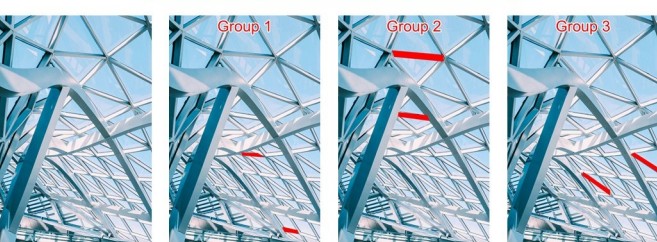

You are a capable vision-based reasoning agent designed to analyze structural connectivity and infer topological relationships between structural members. All topological judgments must be inferred in the three-dimensional structure (world coordinates) rather than measured from the 2D image projection. Your goal is to rank each group based on the topological distance (number of hops) between the two structural members it contains.

**Your Task**
You will be provided with an Original Structure Image and a set of shuffled Annotated Group Images (labeled 1, 2, 3).
Each annotated group contains two structural members, with the visible portions of both members highlighted as red regions. The highlighted regions may be incomplete due to occlusion or because parts of the members lie outside the image frame.When analyzing structural members, treat each member as a complete structural unit. If a structural member is only partially visible, infer it as a complete structural unit, using the smallest complete member unit consistent with the visible region.Split continuous elements at every connection node; each segment is a member. Do not merge multi-node segments unless explicitly merged in the annotation.For each group, determine the topological distance (hops) between the two members, defined as:A hop is one direct connection between two structural members (e.g., physical joint, intersection, or direct attachment).If the two members are directly connected, their topological distance is 1 hop.If the two members are not directly connected, the topological distance is the minimum number of intermediate members required to form a continuous connection path between them.Sort the groups according to their topological distances in ascending order, from smallest number of hops to largest number of hops.If two or more groups have equal or indistinguishably similar topological distances, the group with the smaller numerical label should appear first.

**Output Format**
Your response must be only a Python list of integers representing the order of group labels from smallest topological distance to largest. Do not include any other text, reasoning, or explanation.**Example:** If you determine the order is Group 2 (0 hops), Group 1 (1 hop), Group 3 (2 hops), your output must be:[2, 1, 3]

Now please provide your answer in the requested format.

**A:** [2, 1, 3]

*Figure 19.* Full example for topological-hop distance category.

**Topological - Cycle Length**

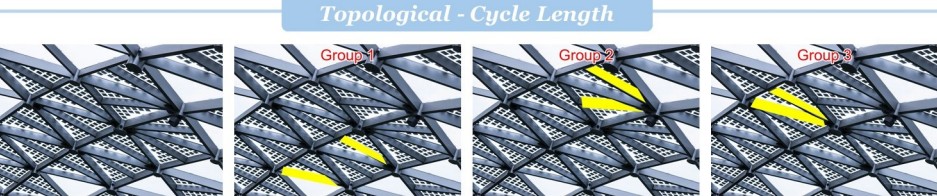

You are a capable vision-based reasoning agent designed to analyze structural connectivity and infer cyclic topological relationships among structural members. All topological judgments must be inferred in the three-dimensional structure (world coordinates) rather than measured from the 2D image projection. Your goal is to rank each group based on the minimum number of edges in a cycle that includes all given members.

**Your Task**
You will be provided with an Original Structure Image and a set of shuffled Annotated Group Images (labeled 1, 2, 3). Each annotated group contains a set of structural members, with the visible portions of all members highlighted as red regions. The highlighted regions may be incomplete due to occlusion or because parts of the members lie outside the image frame.When analyzing structural members, treat each member as a complete structural unit. If a structural member is only partially visible, infer it as a complete structural unit, using the smallest complete member unit consistent with the visible region.Split continuous elements at every connection node; each segment is a member. Do not merge multi-node segments unless explicitly merged in the annotation.For each group, determine the minimum cycle length, defined as:A cycle is a closed topological path formed by connected structural members, where the start and end member coincide.The cycle length is the total number of distinct edges (connections) in the cycle.The cycle must include all members in the group (each member must lie on the cycle).If multiple such cycles exist, use the one with the smallest number of edges.If no cycle exists that includes all given members, treat the cycle length as infinite.Sort the groups according to their minimum cycle lengths in ascending order, from smallest number of edges to largest number of edges.If two or more groups have equal or indistinguishably similar cycle lengths, the group with the smaller numerical label should appear first.

**Output Format**
Your response must be only a Python list of integers representing the order of group labels from smallest cycle length to largest. Do not include any other text, reasoning, or explanation.**Example:** If you determine the order is Group 1 (3 edges), Group 3 (4 edges), Group 2 (no valid cycle), your output must be:[1, 3, 2]

Now please provide your answer in the requested format.

**A:** [3, 2, 1]

*Figure 20.* Full example for topological-cycle length category.

### Topological-Multi View-1

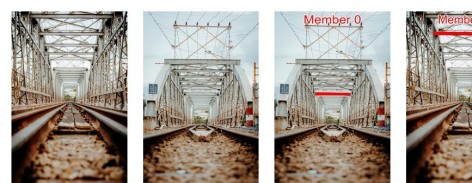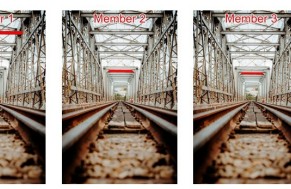

You are a capable vision-based reasoning agent designed to analyze structural connectivity and infer cyclic topological relationships among structural members. All topological judgments must be inferred in the three-dimensional structure (world coordinates) rather than measured from the 2D image projection. Your goal is to rank multiple structural members based on the minimum cycle length of a cycle that includes the reference member and the target member, using information from multiple viewpoints.

**Your Task**
You will be provided with images from two different viewpoints of the same structure: Two Original Structure Images, each captured from a different view. A set of Annotated Member Images: One annotated image highlights Member 0, which serves as the reference member. Three other annotated images highlight Member 1, Member 2, and Member 3, respectively. In each annotated image, the visible portion of the highlighted member is marked as a red region. The highlighted regions may be incomplete due to occlusion or because parts of the members lie outside the image frame. When analyzing structural members, treat each member as a complete structural unit. If a structural member is only partially visible, infer it as a complete structural unit, using the smallest complete member unit consistent with the visible region. Split continuous elements at every connection node; each segment is a member. Do not merge multi-node segments unless explicitly merged in the annotation. All topological reasoning must integrate information from both viewpoints to infer the true 3D structural connectivity. **Cycle Length Definition:** For each of Member 1, Member 2, and Member 3, determine its minimum cycle length with Member 0, defined as: A cycle is a closed topological path formed by connected structural members, where the start and end member coincide. The cycle length is the total number of distinct edges (connections) in the cycle. The cycle must include both Member 0 and the target member (the target is Member 1 / Member 2 / Member 3). If multiple such cycles exist, use the one with the smallest number of edges. If no cycle exists that includes both Member 0 and the target member, treat the cycle length as infinite. **Sorting Requirement:** Sort Member 1, Member 2, and Member 3 according to their minimum cycle lengths with Member 0, in ascending order, from smallest number of edges to largest number of edges. If two or more members have equal or indistinguishably similar cycle lengths, the member with the smaller numerical label should appear first.

**Output Format**
Your response must be only a Python list of integers representing the order of member labels from smallest cycle length to largest.
Do not include any other text, reasoning, or explanation. **Example:** If you determine that the smallest cycle including Member 0 and Member 2 has 3 edges, Member 0 and Member 1 has 4 edges, and Member 0 and Member 3 has no valid cycle, your output must be:[2, 1, 3]

Now please provide your answer in the requested format.

**A:** [3, 2, 1]

*Figure 21.* Full example for topological-multi view-1 category.

### Topological-Multi View-2

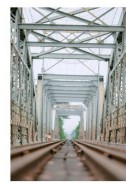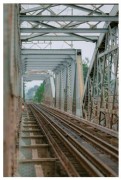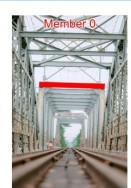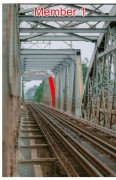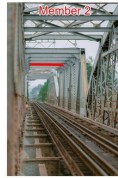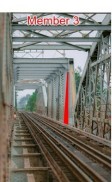

You are a capable vision-based reasoning agent designed to analyze structural connectivity and infer topological relationships between structural members. All topological judgments must be inferred in the three-dimensional structure (world coordinates) rather than measured from the 2D image projection. Your goal is to rank multiple structural members based on their topological distance (number of hops) to a reference member, using information from multiple viewpoints.

**Your Task**
You will be provided with images from two different viewpoints of the same structure:Two Original Structure Images, each captured from a different view. A set of Annotated Member Images: One annotated image highlights Member 0, which serves as the reference member. Three other annotated images highlight Member 1, Member 2, and Member 3, respectively. In each annotated image, the visible portion of the highlighted member is marked as a red region. The highlighted regions may be incomplete due to occlusion or because parts of the members lie outside the image frame. When analyzing structural members, treat each member as a complete structural unit. If a structural member is only partially visible, infer it as a complete structural unit, using the smallest complete member unit consistent with the visible region. Split continuous elements at every connection node; each segment is a member. Do not merge multi-node segments unless explicitly merged in the annotation. **Topological Distance Definition:** For each of Member 1, Member 2, and Member 3, determine its topological distance (number of hops) to Member 0, defined as follows: A hop is one direct connection between two structural members (e.g., physical joint, intersection, or direct attachment). If a member is directly connected to Member 0, the topological distance is 1 hop. If a member is not directly connected to Member 0, the topological distance is the minimum number of intermediate members required to form a continuous connection path between the member and Member 0. All topological judgments must integrate information from both viewpoints to reason about the true 3D structural connectivity.

**Sorting Requirement :** Sort Member 1, Member 2, and Member 3 according to their topological distances to Member 0, in ascending order, from smallest number of hops to largest number of hops. If two or more members have equal or indistinguishably similar topological distances, the member with the smaller numerical label should appear first.

**Output Format**
Your response must be only a Python list of integers representing the order of member labels from smallest topological distance to largest.
Do not include any other text, reasoning, or explanation. **Example:** If you determine that Member 1 is directly connected to Member 0 (1 hop), Member 3 is connected via one intermediate member (2 hops), and Member 2 is farther away (3 hops), your output must be:[1, 3, 2]

Now please provide your answer in the requested format.

**A:** [3, 2, 1]

*Figure 22.* Full example for topological-multi view-2 category.

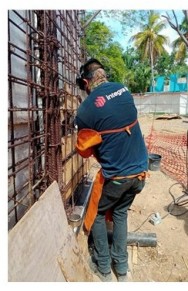 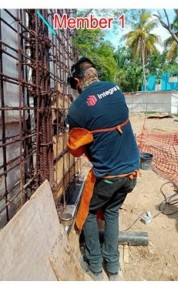 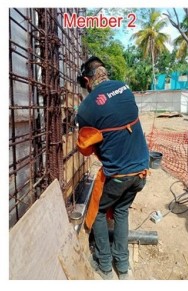 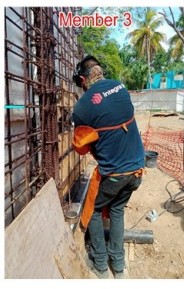 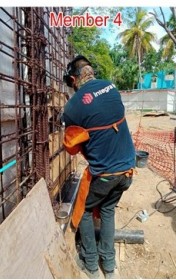

**Question:(Geometric-Ground Height)** You are a capable vision-based reasoning agent designed to analyze structural components and infer spatial relationships from images. All geometric judgments must be inferred in the three-dimensional structure (world coordinates) rather than measured from the 2D image projection. Your goal is to rank structural members based on the heights of their centroids relative to the ground plane......

**Correct Answer:** [4, 2, 3, 1]

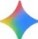 Gemini-3-Pro

Based on the visual analysis of the provided images, the structural members are horizontal rebar ties (stirrups) located at different heights along the vertical reinforcement cage.1.  **Member 4**: The highlighted cyan region is located at the lowest position among the four, roughly aligned with the worker's hip/thigh area and below the horizontal seam of the plywood formwork. This is the lowest member.2. **Member 3**: The highlighted cyan region is located above Member 4, roughly aligned with the worker's waist/elbow area and above the plywood seam.3.  **Member 2**: The highlighted cyan region is located above Member 3, roughly aligned with the worker's shoulder level.4.  **Member 1**: The highlighted cyan region is located at the highest position, near the top of the visible rebar cage, above the worker's head level.Ranking the members by centroid height from lowest to highest:[4, 3, 2, 1]

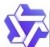 Qwen3-VL-30B-A3B

Got it, let's try to figure out the centroid heights. First, I need to look at each member's cyan highlighted part and infer their 3D positions relative to the ground.

The original structure is a rebar grid. The ground plane is the dirt area. Members are parts of the rebar structure. Let's analyze each:
- Member 1: Cyan line is on the top left part of the rebar. So its visible part is high up.
- Member 2: Cyan line is lower than Member 1, maybe middle height.
- Member 3: Cyan line is even lower, closer to the middle but lower than Member 2?
Wait, no. Wait, the ground is the bottom. So higher members have centroids higher above ground.
Wait, the key is the vertical position (height) of the centroid. Let's think about how the rebar is structured. The rebar grid has vertical and horizontal members. But the problem says "structural members" which are probably the bars. Each annotated image shows a specific bar with cyan.
Looking at the images: Member 1: The cyan highlight is near the top of the rebar structure. So its centroid would be high.Member 2: Cyan highlight is lower than Member 1.Member 3: Cyan highlight is lower than Member 2.Member 4: Cyan highlight is the lowest among them.
Wait, wait. Let's check the positions. In the first image (Original), the rebar is vertical. The ground is the dirt. So the height from ground to the member's centroid.Member 4's cyan line is the lowest (closest to the ground), then Member 3, then Member 2, then Member 1. Wait, no. Wait, in the images: Member 1: cyan is topmost.Member 2: cyan is below Member 1. Member 3: cyan is below Member 2.Member 4: cyan is below Member 3.
So the order from lowest to highest centroid height would be Member 4 (lowest), then Member 3, then Member 2, then Member 1 (highest).Wait, let's confirm. If a member's visible part is lower, its centroid is lower. So:Member 4's cyan is the lowest visible part, so its centroid is lowest.Then Member 3's cyan is next higher.Then Member 2.Then Member 1's cyan is highest.
Yes. So ascending order (lowest to highest) is 4, 3, 2, 1.Wait, let's check again. Let's imagine the rebar structure. The vertical bars: the ones closer to the ground have lower centroids. So if Member 4's highlighted part is the lowest (closest to the ground), then its centroid is lowest. Then Member 3 is next, etc.Yes. So the order should be [4, 3, 2, 1].

*Figure 23.* Illustration of 3D spatial-logic errors (highlighted in blue).

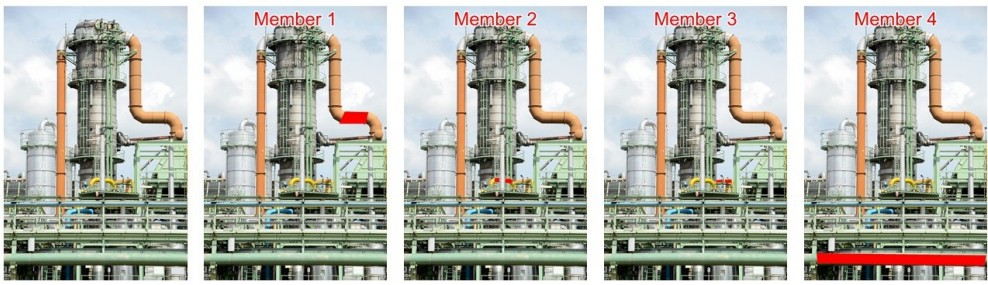

**Question:(Geometric-Ground Height)** You are a capable vision-based reasoning agent designed to analyze structural components and infer spatial relationships from images. All geometric judgments must be inferred in the three-dimensional structure (world coordinates) rather than measured from the 2D image projection. Your goal is to rank structural members based on the heights of their centroids relative to the ground plane......

**Correct Answer:** [4, 2, 3, 1]

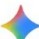 Gemini-3-Pro

Based on the visual analysis of the provided images and the spatial arrangement of the structural members:1. **Member 4**: The red region highlights a large horizontal green pipe located at the very bottom of the image foreground. This is the lowest member relative to the ground.2. **Member 3**: The red region highlights a section of the horizontal blue pipe running along the main pipe rack. This rack is elevated above the ground-level pipe (Member 4) but is situated below the yellow pipe loop.3. **Member 2**: The red region highlights a yellow pipe loop. This loop is supported on a structure that sits above the main pipe rack where Member 3 is located. Therefore, its centroid is higher than that of Member 3.4. **Member 1**: The red region highlights the elbow of a large orange duct located near the top of the central column. This member is positioned significantly higher than the other three pipes.Ranking the members by centroid height from lowest to highest:[4, 3, 2, 1]

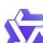 Qwen3-VL-30B-A3B

So, let's try to figure out the centroid heights of each member. First, I need to look at each annotated image and determine where the red region is on the structure, then infer the 3D position.Starting with Member 1: The red region is on the brown pipe that curves around the tall tower. In the original image, that pipe is part of a vertical structure. Wait, but we need to think about the entire member as a complete unit. So for Member 1, the visible red part is on the curved pipe. But how does that relate to height?Member 2: Red region is on the lower part of the green structure (the platform area). So that's probably lower than some others.Member 3: Red region is on the yellow pipes near the middle of the tower. Those are horizontal pipes, so their height would be mid-tower.Member 4: Red region is on the bottom part of the green structure, maybe the base or lower platform.

Wait, no. Wait, the tower is the tallest structure. Let's consider the tower's height. The main tower has multiple levels. The red regions:Member 1: The red is on the large brown pipe that goes around the top part of the tower. So that pipe is high up.Member 2: The red is on the green structure's lower part, maybe the first platform below the tower.Member 3: The red is on the yellow pipes which are on a mid-level platform of the tower.Member 4: The red is on the bottommost part of the green structure, so closest to the ground.

Wait, let's list them by height:Lowest: Member 4 (red at bottom)Next: Member 2 (red on lower platform)Then: Member 3 (red on mid-platform)Highest: Member 1 (red on upper pipe)

Wait, but need to check if that's correct. Let's visualize:Original structure has a tall central tower. The green structures are platforms. The brown pipes are vertical and curved. The silver tank is on the left.Member 4's red is at the bottom of the green structure (so lowest), Member 2's red is on a platform above that (but still low), Member 3's red is on the yellow pipes which are on a higher platform of the tower, Member 1's red is on the curved pipe which is near the top of the tower.Yes, so order from lowest to highest: 4, 2, 3, 1.

Wait, but wait. Let me double-check.Member 4: red region is at the bottom of the green structure. So the centroid of the member (which is the green structure's lower part) is lowest.Member 2: red region is on the green structure's next level up, so higher than Member 4.Member 3: red region is on the yellow pipes, which are on the tower's mid-section. The tower is taller than the green structures, so this is higher than Member 2.Member 1: red region is on the brown pipe that's curving around the top of the tower. So that's the highest.So ascending order (lowest to highest): 4, 2, 3, 1.

Wait, but let's confirm. Are there any other factors? For example, if a member is a pipe that extends vertically, even if only part is visible, we assume it's a complete unit. So Member 1's pipe is part of a vertical structure that goes up, so its centroid is high. Member 4's structure is the lowest part.Yes, so the order should be [4, 2, 3, 1].

*Figure 24.* Illustration of 3D spatial-logic errors (highlighted in blue), Member-extent errors (highlighted in orange), Object-recognition errors (highlighted in red).

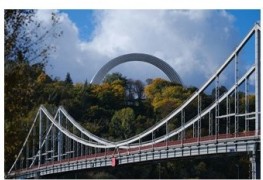 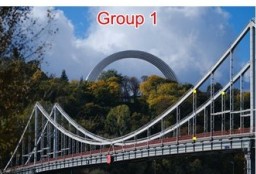 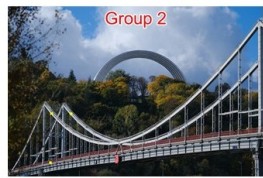 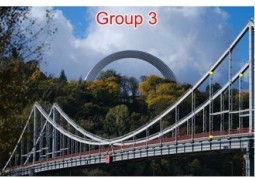

**Question:(Geometric-Area)** You are a capable vision-based reasoning agent designed to analyze structural components and infer geometric properties from images. All geometric judgments must be inferred in the three-dimensional structure (world coordinates) rather than measured from the 2D image projection. Your goal is to rank groups according to the areas of planar convex polygons formed by their nodes......

**Correct Answer:** [ 1, 3, 2 ]

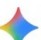 Gemini-3-Pro

Based on the visual analysis of the provided images, we can determine the relative areas of the polygons by estimating the height of the vertical suspender cables defined by the yellow nodes. The width of the polygons (spacing between vertical cables) is constant across the bridge. Therefore, the area is directly proportional to the height of the vertical cables.1. **Group 2**: These nodes are located on the left side of the bridge span, close to the lowest point of the main suspension cable (the center of the span). The vertical cables here are the shortest among the three groups.2. **Group 3**: These nodes are located on the right side of the bridge span, but closer to the center than Group 1. The main cable has begun to rise significantly here, making these vertical cables taller than those in Group 2, but shorter than those in Group 1.3. **Group 1**: These nodes are located on the far right side of the bridge span, where the main suspension cable rises steeply towards the tower. The vertical cables here are the longest, resulting in the polygon with the largest area.
Ranking the groups by area from smallest to largest:[2, 3, 1]

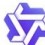 Qwen3-VL-30B-A3B

So, let's try to figure out this problem. First, I need to understand what's being asked. We have three groups (Group 1, 2, 3) each with yellow nodes on the bridge. The task is to rank them by the area of the convex hull formed by their nodes, from smallest to largest.
First, I need to visualize the 3D structure. The bridge is a suspension bridge, so the nodes are points on the bridge structure. The convex hull area depends on how spread out the nodes are in 3D space.Looking at the images: Group 1: Yellow points are probably on the left side of the bridge. Let's check the positions. In Group 1, the yellow nodes might be closer together, forming a smaller polygon.Group 2: The yellow nodes might be more spread out than Group 1 but less than Group 3.Group 3: The yellow nodes are likely the most spread out, so the convex hull area is largest.
Wait, but how do we infer the 3D structure? Since it's a suspension bridge, the main cables form a catenary curve. The nodes are on the bridge deck and the supporting structures.Let me think about the number of nodes and their positions. Each group has yellow points. Let's count the yellow points:Group 1: Let's see the image. Group 1 has yellow points. Maybe two or three? Wait, looking at the images:Group 1 image: The yellow points are on the left part of the bridge. Let's check the positions. The bridge has two main towers (the vertical parts), and the cables go over them. The yellow points for Group 1 are probably on the left cable section.Group 2: The yellow points are maybe on the middle or right?Group 3: The yellow points are more spread out across the bridge.Wait, perhaps Group 1 has nodes that are closer together in 3D space, so their convex hull area is smaller. Group 2 has slightly more spread, Group 3 the most.Alternatively, maybe Group 1 has the smallest area because the nodes are clustered, Group 2 medium, Group 3 largest.
Wait, the problem says "planar convex polygon formed by their nodes". So if the nodes are colinear, the area is zero, but here they are not. So the more spread out the nodes, the larger the area.Looking at the images:Group 1: How many yellow nodes? Let's count. In Group 1 image, there are two yellow dots on the left side. Wait, no, maybe three. Wait, the user provided images:Group 1: The yellow points are on the left part of the bridge. Let's check the original image. The bridge has two main towers. The cables go from the top of the towers down to the deck. The yellow nodes are on the cables.Group 1: Two yellow nodes on the left cable.Group 2: Two yellow nodes on the left cable but maybe further apart.Group 3: Three yellow nodes, spread across the bridge.
Wait, no. Alternatively, the positions of the yellow nodes in 3D:The bridge is a suspension bridge, so the main cables form a parabolic curve (approximate). The nodes on the cables will have different x (horizontal), y (vertical), z (depth) coordinates.Assuming the bridge spans from left to right (x-axis), height (y-axis), and depth (z-axis, into the scene).For Group 1: The yellow nodes are on the left side, so lower x values. If they are close together in x and y, the area is small.Group 2: Nodes are a bit further apart in x.Group 3: Nodes are spread across the bridge, so larger x range, hence larger area.Thus, Group 1 < Group 2 < Group 3.Therefore, the order from smallest to largest is [1, 2, 3].

*Figure 25.* Illustration of 3D spatial-logic errors (highlighted in blue), Computational and comparison-logic errors (highlighted in green).

