# OpenReview forum: "Thinking in Structures: Evaluating Spatial Intelligence in Constraint-Governed Spaces"
_ICML.cc/2026/Conference — ICML 2026 regular_

### Official Review · Reviewer_Qjac · 2026-02-23

**Soundness:** 2
**Presentation:** 3
**Significance:** 2
**Originality:** 3
**Overall Recommendation:** 4
**Confidence:** 4

**Summary:**

This paper introduces SSI-Bench, a new vision-language benchmark designed to evaluate how well models understand complex, real-world 3D engineering structures. Unlike prior benchmarks that focus on unconstrained everyday scenes where models can use 2D shortcuts, SSI-Bench targets Constrained-Manifold Spatial Reasoning (CMSR). In this regime, the physical and geometric rules of structures—like space frames, towers, and bridges—sharply limit the possible 3D configurations, requiring models to move beyond surface-level image correlations. Interestingly, most VLMs fail on this benchmark.

**Compliance With Llm Reviewing Policy:**

Affirmed.

**Final Justification:**

I raised my score because most of my concerns are addressed.

**Key Questions For Authors:**

Given the low accuracy of current models, how do we distinguish between genuine reasoning and lucky guesses in the ~20-30% accuracy range?

Can you provide evidence or a pilot study showing that improvements on SSI-Bench actually correlate with better spatial performance in more general, unconstrained domains?

How do you justify the extreme difficulty of the Area and Volume tasks? How does human perform on these tasks?

**Limitations:**

yes

**Strengths And Weaknesses:**

### Pros

1. The concept of the "Constrained-Manifold" is a mathematically grounded way to frame why current VLMs fail to understand true 3D space.

2. The use of a 400-hour and human-centered annotation pipeline ensures a level of data cleanliness which automated benchmarks often miss.

3. Testing 31 different VLMs provides a clear snapshot of the current state of the field across both proprietary and open-weight models.

### Cons

1. Extreme Task Difficulty and Benchmark Utility:  The performance of even the most advanced VLMs is pretty low, with several models remaining close to the random-ranking baseline of 12.85%. There is a concern that the tasks, particularly regarding Area and Volume calculations on complex structural nodes, are excessively difficult. If the questions are so specialized that a non-expert cannot reliably solve them without ground truth or specific engineering training, the benchmark risks becoming a measure of domain-specific knowledge rather than general spatial intelligence. This "floor effect" limits the benchmark's ability to provide a meaningful signal for model development.

2. Domain Gap and Lack of Generality: The benchmark focuses narrowly on "line segments" and "nodes" within engineering structures. While the authors claim this represents 3D spatial intelligence, there is a significant domain gap between identifying a specific rebar tie in a vertical cage and the "general" spatial performance users actually care about. We typically prioritize a model’s ability to reason about objects and layouts in unconstrained, everyday environments. Success in this niche structural domain does not necessarily translate to improved spatial reasoning in real-world applications like robotics or general scene understanding.

3. Ecological Validity vs. Theoretical Rigor: By moving away from unconstrained scenes, the paper loses ecological validity. The authors argue that unconstrained scenes are "weakly governed" and allow for shortcuts. However, those "shortcuts" (appearance priors and layout correlations) are exactly how humans and practical AI systems operate in the real world. A benchmark that removes these common-sense cues in favor of abstract geometric ranking may be measuring a form of "spatial logic" that is disconnected from the broader goals of multimodal research.

---

> ### Author Rebuttal · Authors · 2026-03-30
>
> We sincerely thank you for the detailed and constructive review. We are pleased that you recognize our high-quality human annotation pipeline and broad model evaluation. Below, we provide new experimental evidence to address your valid concerns regarding task difficulty and generality. New empirical tables and visual comparisons are available in this anonymous document: [**https://anonymous.4open.science/r/ssi-bench/Qjac.pdf**].  All references to "Fig. RX" or "Table RY" point to this supplementary file.
> ### **1. Task Difficulty and Benchmark Utility**
> - **Challenging for VLMs, but not intractable. Table 1** in the main paper shows that SSI-Bench is difficult for current VLMs, but far from “near-impossible”: the random baseline is **12.85%**, while the best VLM (Gemini-3-Flash) reaches **33.6%**. In contrast, human performance is **91.6%**, indicating that the benchmark is hard for models but easy for humans.
> - **Area and Volume are not excessively difficult** (*Q3*). Human accuracy on Area and Volume is **98%** and **91.75%**, respectively (**Table 1**). During construction, we typically ensured that candidate components share the same base length or area (**Fig. R1**), so these questions mainly reduce to intuitive 3D height comparison rather than advanced engineering computation.
> - **Non-expert humans also perform strongly.** We conducted a supplementary study with lay participants without structural-engineering background. After a brief tutorial (**<10 minutes**), they achieved **91%** accuracy (**Table R1**), nearly identical to the **92%** achieved by participants with basic engineering knowledge.
> - **Difficult, but still highly discriminative.** We appreciate the concern about a possible floor effect. However, if SSI-Bench truly suffered from a floor effect, we would not expect stable separation across models. Instead, we observe: (i) consistent gaps between proprietary and open-source models; (ii) generational improvements within model families; and (iii) distinct task-wise performance profiles. Thus, SSI-Bench is not random noise, but a discriminative benchmark in a low-accuracy regime.
> - **Genuine reasoning vs. lucky guesses** (*Q1*). Lucky guessing is inherently unlikely for ranking tasks, whose answer space is much larger than that of standard multiple-choice or binary questions. To further examine this, we analyzed cases where Gemini-3-Pro was correct but Qwen3-VL-8B was wrong. Among 217 such cases, we randomly inspected **30** and found that **26** reflected genuine reasoning rather than lucky guesses. Moreover, the main paper reports pairwise accuracy (**Table 4**), which measures whether candidate pairs are ordered correctly (e.g., Gemini-3-Flash = **66.7%**), providing additional evidence of partial understanding beyond exact full-ranking accuracy.
> ### **2. Domain Gap and Generality**
> We wish to clarify that SSI-Bench is not restricted to a narrow industrial domain. Instead, it serves as a valuable **structure-centric** complement to **scene-centric** unconstrained spatial benchmarks, evaluating a distinct dimension of spatial intelligence.
> - **Non-Specialized of Data Sources:** The images in our benchmark are exclusively sourced from **open-access daily photography platforms** (e.g., Pexels). They reflect authentic, real-world urban scenes captured in everyday settings, not specialized industrial blueprints. Highly specialized images, such as the "vertical rebar cages" you mentioned, actually account for **less than 5%** of the dataset.
> - **High Correlation with General Spatial Abilities** (*Q2*): To answer your question of whether improvements on SSI-Bench translate to better spatial performance in more general domains, we analyzed the consistency between model rankings on SSI-Bench and those on other mainstream, unconstrained spatial benchmarks. As shown in **Table R2**, they exhibit a highly positive correlation. This strongly demonstrates that SSI-Bench evaluates fundamental spatial intelligence rather than specialized knowledge in structural engineering.
> ### **3. Ecological Validity vs. "Shortcuts"**
> We strongly agree that humans and AI rely on prior knowledge (appearance and layout correlations) in the real world. However, **the "shortcuts" we aim to eliminate are strictly "2D pixel artifacts"** (e.g., pixel height), not common sense itself. In fact, the structural constraints evaluated in SSI-Bench embody the exact "3D common-sense priors" humans use to parse complex environments. Therefore, SSI-Bench fully retains ecological validity: it simply forces models to abandon 2D heuristics and utilize genuine 3D physical priors (**Fig. R2**), just as humans do.
>
> We are more than happy to integrate the above discussions and experiments into the final version of the paper. We hope this new evidence thoroughly addresses your concerns.

---

> > ### Author Rebuttal · Reviewer_Qjac · 2026-04-02
> >
> > Thank you for the detailed rebuttal. I agree that the additional results help address the strongest version of my “floor effect” concern: the benchmark is clearly difficult for current VLMs, but not intractable, and the reported model separation suggests it is not pure noise.
> >
> > I also find the clarification on Area/Volume questions helpful, especially if the lay-participant study is included in the final paper, since that would support the claim that the benchmark tests spatial reasoning rather than specialized engineering knowledge.
> >
> > I remain partially convinced about the broader claims of generality: the key issue is not whether the images come from daily photography platforms, but whether a structure-centric task distribution can meaningfully represent the wider spatial capabilities users care about in unconstrained environments. Though reducing 2D shortcuts is a reasonable design choice, it also introduces a trade-off by making the benchmark less ecologically representative of real-world visual reasoning, where appearance and layout priors are naturally used.

---

> > > ### Author Response · Authors · 2026-04-02
> > >
> > > Thank you very much for your timely follow-up, your constructive feedback, and for acknowledging our efforts and clarifications in the rebuttal phase. We deeply appreciate your open-mindedness and the time you have dedicated to helping us improve this work. We will ensure that all the supplementary experiments provided during the rebuttal are incorporated into the final version of the manuscript.
> > >
> > > Regarding your remaining valid concern about generality and ecological validity, we agree with your insightful observation. In everyday, unconstrained environments, humans and practical AI systems naturally and necessarily rely on 2D appearance and layout priors.
> > >
> > > Following your feedback, we have deeply reflected on the positioning of our benchmark and would like to clarify our perspective through the following points:
> > > ### **1. A Diverse Evaluation Ecosystem**
> > > We firmly believe that evaluating foundational models requires a diverse, multi-dimensional ecosystem of benchmarks. SSI-Bench is not designed to replace the existing, rich landscape of scene-centric spatial benchmarks. Those benchmarks are already comprehensive and essential. Instead, SSI-Bench aims to provide a currently missing perspective: an isolated evaluation of the underlying 3D logical engine.
> > > ### **2. The Unique Ecological Niche of SSI-Bench**
> > > We view SSI-Bench as occupying a unique and necessary ecological niche—a pragmatic middle ground between real-world comprehensive reasoning and purely abstract spatial reasoning.
> > > - On one end of the spectrum, benchmarks like VSI-Bench [1] or MMSI-Bench [2] directly serve the broad, comprehensive spatial needs of embodied AI in everyday scenes.
> > > - On the other extreme, benchmarks like Spatial457 [3] or specific tasks in OmniSpatial [4] focus on pure spatial reasoning, operating largely independently of real-world contexts (e.g., using purely abstract mathematical shapes/synthetic objects). We have updated the anonymous link [https://anonymous.4open.science/r/ssi-bench/Qjac.pdf ]; please refer to **Fig. R3** for visual examples of this category.
> > >
> > > As you aptly noted, SSI-Bench is a deliberate trade-off. We deliberately select appropriate "pure 3D" structures from the real world to conduct a "stress test" on the foundational 3D logic of models. It sacrifices some everyday visual priors to ensure that the measured 3D reasoning is unpolluted by 2D shortcuts. Nevertheless, in doing so, it successfully strikes a necessary balance between real-world authenticity and the purity of 3D reasoning.
> > >
> > > *[1] Yang, Jihan, et al. "Thinking in space: How multimodal large language models see, remember, and recall spaces." CVPR (2025).*
> > >
> > > *[2] Yang, Sihan, et al. "Mmsi-bench: A benchmark for multi-image spatial intelligence." ICLR (2026).*
> > >
> > > *[3] Wang, Xingrui, et al. "Spatial457: A diagnostic benchmark for 6d spatial reasoning of large mutimodal models." CVPR (2025).*
> > >
> > > *[4] Jia, Mengdi, et al. "Omnispatial: Towards comprehensive spatial reasoning benchmark for vision language models." ICLR (2026).*
> > > ### **3. Enriching the Discussion in the Camera-Ready Version**
> > > In the camera-ready version, we will explicitly include this discussion to honestly articulate the deliberate trade-off we made by removing 2D shortcuts to isolate pure 3D reasoning. This will help readers clearly understand the exact boundary conditions and the unique ecological niche of SSI-Bench, without over-generalizing its scope.
> > >
> > > Thank you again for your rigorous inquiry into these fundamental issues. Your feedback prompted us to deeply reflect on the true significance of SSI-Bench. Through this constructive debate, our understanding of the benchmark's exact positioning has crystallized. We truly appreciate the time and effort you've invested in improving our work.

---

### Official Review · Reviewer_AQZ6 · 2026-03-05

**Soundness:** 2
**Presentation:** 3
**Significance:** 3
**Originality:** 3
**Overall Recommendation:** 4
**Confidence:** 3

**Summary:**

The paper introduces SSI-Bench, a human-curated benchmark of 1,000 ranking-based VQA questions on complex real-world 3D engineering structures to evaluate constrained (geometry/topology/feasibility) spatial reasoning, including both single- and multi-view settings. It further provides a large-scale evaluation of 31 VLMs plus humans and an error analysis showing a substantial human–model gap.

**Compliance With Llm Reviewing Policy:**

Affirmed.

**Final Justification:**

The author has addressed all my concerns. I believe the quality of the benchmark is satisfactory. Therefore, I am inclined to accept this paper.

**Key Questions For Authors:**

1. **Evidence for issues in prior benchmarks (lines 71–79):** Can you add concrete, image-based examples showing the shortcut cues in representative prior benchmarks, and then show analogous examples where SSI-Bench avoids them?


2. **Are SSI-Bench questions truly requiring 3D reasoning vs. 2D cues (especially the teaser):** Do you have any analyses/ablations to quantify how well simple 2D heuristics (e.g., pixel height, bbox size, 2D angle, vertical position) can solve each task type?


3. **Clarification/justification of the term “constrained manifold”:** Would you consider renaming it or adding a formal clarification to avoid mathematical overclaiming?

**Limitations:**

The authors discuss limitations, but they do not sufficiently discuss potential negative societal impacts. Stronger structural/spatial reasoning could enable or improve dual-use applications (e.g., surveillance, military/targeting) and that the benchmark could be used to optimize models for such purposes.

**Strengths And Weaknesses:**

## Strengths

- **Presentation:** The writing and figures are clear, and the logic/organization is good.
- **Originality and Significance:** The paper points out potential issues in prior spatial reasoning benchmarks, and uses constrained structural settings to address these issues.
- **Soundness:** The dataset is human-annotated and checked, so the data quality is high. The task types are comprehensive and diverse. The paper evaluates 31 VLMs, so the experiments are thorough.

## Weaknesses

- **Soundness and Presentation:** The potential issues in prior spatial reasoning benchmarks (lines 71–79 in the paper) should be demonstrated with more images to show these issues and why constraints can address them; explaining only in text is not intuitive enough.
- The paper claims “spatial reasoning on complex real-world 3D structures,” but the teaser question seems answerable using only 2D information.
- I think the paper overuses/misuses the term “manifold,” which is mathematically very imprecise.

---

> ### Author Rebuttal · Authors · 2026-03-30
>
> We sincerely thank you for your detailed feedback and for recognizing our clear presentation, thorough evaluation, and high data quality. Your constructive suggestions have significantly strengthened our paper. Below, we address your concerns point-by-point. New empirical tables and visual comparisons are provided in this anonymous document: [**https://anonymous.4open.science/r/ssi-bench/AQZ6.pdf**]. All mentions of "Fig. RX" or "Table RY" refer to this document.
> ### **1. Visual Evidence for Issues in Prior Benchmarks (Lines 71–79)**
> As you insightfully pointed out, text alone is often insufficiently intuitive. To concretely illustrate the limitations of existing benchmarks and how structural constraints overcome them, we present detailed image-based comparisons:
> - **Avoiding 2D Shortcuts:** In most previous spatial benchmarks, the AI acts as a participant in everyday scenes, focusing on general locations and orientations, which provides potential loopholes for 2D shortcuts. In contrast, SSI-Bench is strictly structure-centric. Here, the AI acts as an observer focusing on 3D geometry and topology governed by structural constraints. During annotation, we deliberately select candidate members that circumvent 2D visual shortcuts. Furthermore, the vast number of permutations inherent in strict ranking tasks effectively precludes the possibility of models guessing the correct answer via superficial cues. In **Fig. R1**, we provide visual comparisons between SSI-Bench and prior benchmarks.
> - **Avoiding Ambiguous Answers:** Existing benchmarks generally lack rigid 3D constraints, leaving 3D states highly under-determined from 2D projections alone, which is why strict ranking formulations are rarely adopted in prior works. Rigid structural constraints guarantee that valid 3D configurations concentrate on constrained states, ensuring 3D spatial relations can be deterministically inferred from single 2D views. **Fig. R2** visually contrasts this to show how SSI-Bench resolves ambiguity.
> ### **2. Teaser Question & 2D Heuristics Ablation**
> - **The Teaser Image:** At first glance, the teaser question might appear solvable using only 2D cues, but it is actually designed as a clear example of perspective illusion. From the 2D pixel layout alone, Member 1 appears "higher" than Member 4. However, understanding the 3D structure reveals that Member 4 is a horizontal top beam, while Member 1 is a diagonal brace beneath it. Only by reasoning about these specific structural constraints can a model correctly deduce that Member 1 is physically lower. We detail this reasoning process in **Fig. R3**.
> - **2D Heuristics Ablation:** Following your valuable suggestion, we quantified the effectiveness of simple 2D heuristics. Using the pixel coordinate metadata we recorded during our annotation process, we constructed baselines based entirely on 2D cues (e.g., pixel height, 2D angle, and 2D area; see **Table R1**). The results show that these 2D heuristics perform extremely poorly (**Acc=7.8%**). This analysis strongly supports our claim that SSI-Bench effectively circumvents 2D shortcuts, requiring models to perform genuine 3D spatial reasoning.
> ### **3. Modifications Regarding "Constrained Manifold"**
> We appreciate your rigorous attention to terminology. To avoid potential misunderstandings or mathematical overclaiming regarding the term "manifold," we will rename this concept to the more intuitive **"Structural Constraints"** throughout the revision. Furthermore, we have now formally categorized these constraints into three main types: Geometric Regularity, Topological & Connectivity, and Physical & Semantic (**Table R2**). We have provided annotations of the specific constraint types for the samples in SSI-Bench. These updates eliminate imprecise phrasing and instantiate exactly what these physical and geometric rules are.
> ### **4. Potential Negative Societal Impacts**
> We highly appreciate your foresight regarding the dual-use nature of spatial intelligence. You are entirely correct that stronger structural/spatial reasoning could potentially be optimized for dual-use applications (e.g., surveillance, military/targeting). In our revision, we will add a dedicated **"Broader Impacts"** section to explicitly acknowledge and thoroughly discuss these risks. However, we believe that establishing rigorous evaluation paradigms is a necessary first step toward building more robust, interpretable, and safe spatial AI, which will ultimately yield greater positive impacts for the community.
> We hope these concrete visual examples and quantitative ablations effectively address your concerns, and we will seamlessly integrate them into the final version.

---

> > ### Author Rebuttal · Reviewer_AQZ6 · 2026-04-01
> >
> > Thank you for your detailed reply. My concern has been resolved, so I will raise my score.

---

> > > ### Author Response · Authors · 2026-04-02
> > >
> > > Thank you very much for your timely reply and for your positive feedback on our rebuttal. We are glad to hear that our responses and the additional experiments effectively addressed your concerns.
> > >
> > > We have noted your supportive comment regarding the score adjustment. We truly appreciate your recognition of the merits of SSI-Bench and your constructive suggestions, which have helped us further strengthen the paper.

---

### Official Review · Reviewer_UPqF · 2026-03-06

**Soundness:** 2
**Presentation:** 4
**Significance:** 3
**Originality:** 2
**Overall Recommendation:** 4
**Confidence:** 3

**Summary:**

This paper introduces SSI-Bench, a structure-centric VQA benchmark of 1,000 multiple-choice ranking questions (each with 3–4 candidates) spanning two task families (Geometric and Topological) and an additional Multi-View subset. The benchmark is built via a fully human-centered pipeline with independent reviewers for quality control. The authors evaluate 31 VLMs under a standardized setup and report a large gap between current models and human performance.

**Compliance With Llm Reviewing Policy:**

Affirmed.

**Final Justification:**

I raised my score because the rebuttal addressed most of my concerns. My main remaining concern is that the structural constraints are only implicitly invoked.

**Key Questions For Authors:**

Can you clarify what you mean by “Candidate sets are curated so that correct ordering is not reliably recoverable from 2D pixel layouts alone, but instead requires reasoning about the underlying 3D structure and manifold-induced constraints.”?

How can one evaluate whether the model fail due to violating the manifold constraint vs some other error?

Have you considered automating parts of the generation process?

How was the 512 pixel decided, would it be too small in certain cases where the candidates are harder to tell? Were the humans also tested on the resized images?

**Limitations:**

yes

**Strengths And Weaknesses:**

Strengths

- SSI-Bench is built through a fully human process, and each question undergoes independent checking.
- Broad, standardized model evaluation. The paper evaluates 31 VLMs with temperature 0 and consistent evaluation. Human comparisons is also included.
- The paper is easy to read and follow with clear visuals.

Weaknesses
- The “constrained manifold” contribution is unclear. The paper defines a feasible set M using equality and inequality constraints. However, the paper does not instantiate these constraints for benchmark instances (i.e., it does not specify concrete forms of c and h for the dataset), and the actual data instances are only question/answer with images without constraints. As a result, it is unclear how directly the benchmark tests reasoning under “manifold constraints,” versus testing spatial ranking performance more broadly.
- Scalability. The authors emphasize that the construction process is fully human-centered, and acknowledge concerns about scaling to larger datasets or generating continually fresh test sets.  However, some parts of the benchmark might be extensible in a semi-automatic way—for example, once per-candidate metadata (height, angle, length, etc.) is annotated, one could generate additional ranking questions by sampling subsets of candidates from a larger pool within the same image. This could increase coverage beyond 3–4 candidates and reduce manual effort per instance.

Minor comment:
Terminology consistency. The paper uses phrases such as “constrained manifold” and “manifold constraints.” It would help to define the terminology once and use it consistently, to avoid ambiguity about whether these refer to the same concept.

---

> ### Author Rebuttal · Authors · 2026-03-30
>
> We really appreciate your constructive feedback and your recognition of our fully human-centered pipeline, standardized evaluation, and clear presentation. Your insights are invaluable for improving this work. We address your questions point-by-point below. New empirical tables and visual comparisons are provided in this anonymous document: [**https://anonymous.4open.science/r/ssi-bench/UPqF.pdf**]. All mentions of "Fig. RX" or "Table RY" refer to this document.
>
> ### **1. Modifications and Clarifications Regarding "Constrained Manifold"**
> - **Terminology Revision:** To avoid ambiguity and improve readability, we will replace the term "manifold" with the more intuitive **"Structural Constraints"** throughout our revision.
> - **Categorization and Instantiation:** We have categorized these constraints into three primary types: Geometric Regularity, Topological & Connectivity, and Physical & Semantic (**Table R1**). These specific categories serve as the concrete instantiations of the mathematical equations $c$ and $h$ for our dataset. We have annotated these types for all samples in SSI-Bench. Complete annotations will be included in the final public release.
> - **The Role of Structural Constraints:** While these constraints are the foundational motivation for SSI-Bench, they act as implicit underlying rules guiding both question formulation and required reasoning, rather than explicit prompt inputs. To illustrate why SSI-Bench specifically evaluates reasoning under structural constraints—rather than broad spatial ranking—we compare it with existing benchmarks: (1) **Structure-Centric vs. Scene-Centric:** Unlike prior benchmarks focusing on general spatial properties, SSI-Bench is strictly structure-centric. VLMs must analyze underlying structural constraints, making the comprehension of these rules a prerequisite for correct reasoning. (2) **Ensuring Unambiguous Answers:** These constraints ensure deterministic 3D states, enabling reliable 3D spatial ranking from single 2D views. Lacking these constraints, existing benchmarks leave 3D states highly under-determined from 2D projections. (**Fig. R1**).
> ### **2. Scalability & Automation**
> We highly appreciate your insightful suggestion regarding semi-automatic expansion. Following your advice, we have successfully implemented this approach: by annotating per-candidate metadata and automatically sampling combinations, we can combinatorially scale up the question count with minimal manual effort. New experiments (**Table R2**) demonstrate that this semi-automatic pipeline is highly feasible and consistently generates high-quality test sets. Furthermore, existing ranking tasks can be **effortlessly decomposed** into multiple-choice or binary formats to further expand the scale and diversity of the benchmark.
> ### **3. Clarifying 2D Pixel vs. 3D Structure**
> Due to perspective illusions, 2D pixel distances often deceive the observer. For instance, two components appearing adjacent in the 2D plane might actually be far apart in 3D depth. Relying solely on 2D visual heuristics inevitably leads to incorrect rankings. Only by identifying structural constraints (e.g., recognizing whether members are horizontal or vertical in 3D space) can models break this 2D visual illusion and infer the true 3D physical relations. We provide visual examples (**Fig. R2**) to intuitively illustrate this necessity.
> ### **4. Error Attribution and Failure Analysis**
> As shown in the error analysis in Section 4.4 of the manuscript, we identified model failure modes through sentence-by-sentence analysis of VLM reasoning traces. The four primary error categories all fall under the umbrella of structural constraint recognition errors. We quantified their proportions and found that these constraint-related errors account for **97%** of total failures. This demonstrates that, in SSI-Bench, model failures are overwhelmingly driven by an inability to understand the underlying structural constraints.
> ### **5. The 512-pixel Decision & Human Testing**
> We adopted 512-pixel inputs for VLMs primarily to control token costs and align with prior benchmarks [1]. This resolution is visually sufficient for SSI-Bench. To address your concern, we evaluated additional human volunteers on a 100-sample subset resized to 512 pixels (**Table R3**). Human accuracy remained exceptionally high (showing only a marginal drop **from 92% to 90%**). This definitively proves that 512 pixels preserve adequate details for structural reasoning. The massive human-model gap thus stems entirely from models' spatial reasoning bottlenecks rather than resolution limits.
>
> *[1] Wang, Qineng, et al. "Enact: Evaluating embodied cognition with world modeling of egocentric interaction." ICLR (2026).*
>
> We hope our responses and the new empirical evidence effectively address your concerns. We will incorporate all these discussions into the final version.

---

> > ### Author Rebuttal · Reviewer_UPqF · 2026-04-02
> >
> > Thank you for the clarifications and additional experiments. Most of my concerns have been addressed, and I am therefore raising my score. That said, I am still not fully convinced that the structural constraints, since they are only implicitly invoked in the reasoning process, cannot sometimes be bypassed through other cues or heuristics rather than genuine constraint-aware reasoning.

---

> > > ### Author Response · Authors · 2026-04-03
> > >
> > > Thank you very much for your time reviewing our rebuttal, acknowledging our efforts, and raising your score. We deeply appreciate your constructive engagement throughout this process.
> > >
> > > Your lingering concern regarding whether models might bypass genuine constraint-aware reasoning via alternative heuristics or spurious cues is highly insightful. We completely agree that this is a fundamental and pervasive challenge in VLM evaluation.
> > >
> > > To directly addresås this, we would like to refer you to an additional quantitative ablation we conducted and detailed in our **response to Reviewer AQZ6**. We completely agree that we must prove models cannot simply "game" the benchmark using superficial visual cues. Therefore, we constructed heuristic baselines relying purely on 2D visual shortcuts (e.g., relying solely on 2D pixel height, 2D angle, and 2D area).
> > >
> > > As shown in **Table R1** of the anonymous PDF we prepared for Reviewer AQZ6 (link: [https://anonymous.4open.science/r/ssi-bench/AQZ6.pdf ]), these 2D heuristics perform extremely poorly, achieving an accuracy of **only 7.8%**. This extremely low performance of heuristic baselines, combined with our error analysis (where **97%** of current VLM failures are tied to structural constraint-recognition errors), strongly indicates that SSI-Bench effectively circumvents common 2D shortcuts. The vast permutations in our multiple-choice ranking tasks actively penalize superficial guessing, forcing the models to perform genuine 3D structural reasoning.
> > >
> > > However, your point remains extremely valid for the future trajectory of spatial AI: as models grow more capable, they might develop more sophisticated, non-transparent shortcuts. To ensure this important nuance is communicated, we will explicitly add a discussion on this limitation in our final manuscript. We will cite this 7.8% heuristic ablation to demonstrate how we currently mitigate it, while acknowledging that completely ruling out all implicit shortcut learning remains an open challenge for the community.
> > >
> > > Thank you again for helping us make this paper more rigorous and comprehensive!

---

### Official Review · Reviewer_TQaU · 2026-03-15

**Soundness:** 4
**Presentation:** 4
**Significance:** 3
**Originality:** 3
**Overall Recommendation:** 4
**Confidence:** 3

**Summary:**

The paper introduces SSI-Bench, a new VQA benchmark designed to evaluate "Constrained-Manifold Spatial Reasoning" (CMSR). Unlike existing benchmarks that focus on unconstrained everyday scenes, SSI-Bench uses 1,000 human-annotated ranking questions centered on complex 3D engineering structures (e.g., towers, bridges, scaffolds). These structures are governed by strict geometric and physical laws, making spatial relations more stable and quantifiable. The authors evaluate 31 VLMs, revealing a massive performance gap: the best model (Gemini-3-Flash) achieves 33.6% accuracy compared to 91.6% for humans.

**Compliance With Llm Reviewing Policy:**

Affirmed.

**Key Questions For Authors:**

1. Tie-Breaking Fairness: For symmetric or visually identical components (e.g., Figure 2, Dimension 2 & 3), does the "smaller index first" rule penalize models that correctly perceive them as equal? Would a "tie-aware" metric be more appropriate?

2. How does SSI-Bench fundamentally differ from existing relative spatial benchmarks including 3D relative position (like 3DSRbench) and 3D orientation[1] and the same "constrained manifold" value be achieved by simply re-annotating datasets (like Spatialrgpt-bench) into a ranking format?

[1] Wang, X. et al. "Spatial457: A diagnostic benchmark for 6d spatial reasoning of large mutimodal models." Proceedings of the Computer Vision and Pattern Recognition Conference. 2025.

**Limitations:**

No.

**Strengths And Weaknesses:**

# Strengths

1. Benchmark quality: High-Quality Annotation: Employs a rigorous, human-centered pipeline (400+ hours) with independent cross-verification to ensure tasks are unambiguous and non-trivial. Uses a ranking formulation ($K=3,4$) that provides a more precise measurement of comparative reasoning than binary or open-ended questions.

2. Provides a detailed error analysis (3D spatial-logic, member-extent, etc.) and explores the impact of "thinking" tokens on performance.

# Weakness:

1. Limited Scale and Diversity: Because the dataset depends entirely on manual work, it is relatively small. It is also limited to the narrow field of architecture and construction, which may not show how well models handle general, everyday spatial reasoning.

2. Visual Ambiguity: Some tasks in Figure 2 (such as ground height and dimensions) appear to be "ties" in 3D space. Without 3D ground truth (like CAD models), it is unclear if the "correct" ranking is truly objective or just a subjective guess by the annotators.

---

> ### Author Rebuttal · Authors · 2026-03-30
>
> We sincerely appreciate your insightful and professional comments. We are delighted by your recognition of the high quality of SSI-Bench and our detailed error analysis. Your encouraging feedback **means a lot to us and motivates us** to continue refining this line of research.
>
> We have carefully considered your constructive feedback and address your questions point-by-point below. To support our responses, we have provided new empirical tables and visual comparisons in the following anonymous document: [**https://anonymous.4open.science/r/ssi-bench/TQaU.pdf**]. Please note that all mentions of 'Fig. RX' or 'Table RY' refer to this supplementary material.
>
> ### **1. Scale and Diversity**
> - **Scale:** We acknowledge that the initial dataset size is constrained by manual annotation costs. Inspired by Reviewer UPqF, we have conducted supplementary experiments demonstrating a semi-automated expansion pipeline. By annotating extra structural members in existing images and automatically sampling their combinations, we rapidly generated 1,000 new questions from 100 images with minimal additional manual labor. Importantly, evaluations on this expanded set yield performance comparisons highly consistent with the original benchmark  (**Table R1**). Additionally, the scale of tasks can be further expanded by automatically decomposing ranking questions into multiple-choice or binary formats.
> - **Diversity:** Rather than specialized industrial blueprints, SSI-Bench features ubiquitous everyday structures (e.g., roofs, stairs, bridges) from open platforms like Pexels. Furthermore, our added supplementary analysis reveals a strong positive correlation between model performance on SSI-Bench and general spatial benchmarks (**Table R2**). This demonstrates that despite its unique focus on 3D structural constraints, SSI-Bench effectively evaluates general spatial reasoning capabilities.
> ### **2. Visual Ambiguity**
> - **Rigorous Handling of "Ties":** We clarify that ties are not accidental ambiguities, but anticipated cases systematically addressed during benchmark creation. To eliminate ambiguity, we enforce a deterministic rule: “place the smaller index first.” Crucially, this is not merely for human annotators; **VLMs are also explicitly informed of this rule**. (Due to space limits, the full VLM prompts are detailed in Appendix F).
> - **Objectivity without 3D CAD:** Regarding the absence of CAD models, the objectivity of our ground truth is guaranteed through two mechanisms: (1) **Constraint-Driven Uniqueness:** The strict physical rules governing structures act as an implicit ground truth. These constraints generally guarantee unique 3D relationships that can be reliably inferred from 2D perspectives (detailed in **Part 4** below). (2) **Strict Human Validation:** Whenever structural constraints were insufficient to ensure uniqueness, the task was revised or excluded. Concretely, every task underwent a strict multi-round review involving **at least three** independent individuals. This high consensus ensures our answers reflect objective reality.
> ### **3. Rationale for Tie-Breaking Fairness**
> Our deterministic rule ("smaller index first") does not penalize models; rather, it enhances evaluation rigor. Because models are explicitly prompted with this tie-breaking instruction, strictly enforcing it ensures a precise assessment of their true capabilities. Adopting a lenient "tie-aware" metric (e.g., accepting both A < B and B < A when A = B) would inadvertently **reward arbitrary guessing**, masking whether a model genuinely perceived the spatial equality or simply guessed.
> ### **4. Difference from Existing Benchmarks**
> SSI-Bench cannot be simply replicated by re-annotating existing datasets due to a fundamental difference: **structural constraints**. Unlike prior scene-centric benchmarks, SSI-Bench is strictly structure-centric (**Fig. R1**). Its rigid constraints (e.g., Geometric Regularity, Topological & Connectivity, and Physical & Semantic) restrict valid configurations to deterministic 3D states. This ensures 3D spatial relations can be reliably inferred from single 2D views, enabling strict ranking tasks. Existing benchmarks lack these constraints, leaving 3D states highly under-determined from 2D projections. As a supplement to highlight this orthogonality, we compared SSI-Bench with your mentioned benchmarks (**Fig. R2**). We will emphasize this uniqueness in the revision.
>
> We hope our responses have effectively addressed your concerns and we are more than happy to include all these discussions in the camera-ready version of this work.

---

> > ### Author Rebuttal · Reviewer_TQaU · 2026-04-05
> >
> > Thank you for the detailed rebuttal. The clarifications provided, particularly regarding the expansion pipeline and the handling of visual ambiguities, are very helpful and adequately address my main concerns. I think the paper is in good shape, and I will maintain my Weak Accept score.

---

> > > ### Author Response · Authors · 2026-04-05
> > >
> > > Thank you very much for reviewing our rebuttal and for your positive acknowledgment. We deeply appreciate the time and effort you have dedicated to the review process. Your insightful and constructive feedback has been invaluable in helping us refine and improve our work.

---

### Decision · Program_Chairs · 2026-04-30

**Decision:**

Accept (regular)

**Comment:**

The paper introduces SSI-Bench, a human-curated VQA benchmark targeting spatial reasoning under structural constraints using complex  engineering-like structures. The dataset is carefully constructed via a fully human-centered pipeline and 31 VLMs and humans are evaluated, revealing a human-model gap. All reviewers agree that this benchmark is valuable, and I agree. There are remaining concerns around scale and diversity, which the authors should try and address in the revised version.